# Metrics Matter: A Closer Look on Self-Paced Reinforcement Learning

## Abstract

Curriculum reinforcement learning (CRL) allows to solve complex tasks by generating a tailored sequence of learning tasks, starting from easy ones and subsequently increasing their difficulty. However, the generation of such task sequences is largely governed by application assumptions, often preventing a theoretical investigation of existing approaches. Recently, Klink et al. (2021) showed how self-paced learning induces a principled interpolation between task distributions in the context of RL, resulting in high learning performance. So far, this interpolation is unfortunately limited to Gaussian distributions. Here, we show that on one side, this parametric restriction is insufficient in many learning cases but that on the other, the interpolation of self-paced RL (SPRL) can be degenerate when not restricted to this parametric form. We show that the introduction of concepts from optimal transport into SPRL prevents aforementioned issues. Experiments demonstrate that the resulting introduction of metric structure into the curriculum allows for a well-behaving non-parametric version of SPRL that leads to stable learning performance across tasks.

## 1 Introduction

Reinforcement learning (RL) (Sutton & Barto, 1998) has celebrated great successes as a framework for autonomous acquisition of desired behavior. With ever-increasing computational power, this framework and the algorithms developed under it have allowed to create learning agents capable of solving non-trivial long-horizon planning (Mnih et al., 2015; Silver et al., 2017) and control tasks (Akkaya et al., 2019). However, these successes have also highlighted the need for certain forms of regularization, such as leagues in the context of boardgames (Silver et al., 2017), a gradual diversification of simulated training environments for robotic manipulation (Akkaya et al., 2019) or a tailored training pipeline in the context of humanoid control for soccer (Liu et al., 2021). These regularizations help to overcome shortcomings of modern RL agents, such as poor exploratory behavior – a problem that is an active topic of research (Bellemare et al., 2016; Ghavamzadeh et al., 2015; Machado et al., 2020).

One can view aforementioned regularizations under the umbrella term of curriculum reinforcement learning (Narvekar et al., 2020), where the idea is to avoid shortcomings of modern (deep) RL agents such as aforementioned poor exploration by learning on a tailored sequence of tasks. Such curricula can materialize in a variety of ways and are motivated from many perspectives in the literature (Andrychowicz et al., 2017; Florensa et al., 2017; Wöhlke et al., 2020). Although the resulting curricula can often be interpreted as a sequence of task distributions, these sequences typically lack a formal connection to the reinforcement learning objective of maximizing the expected reward under a given target task distribution. In a recent line of work, Klink et al. (2021) proposed the idea of self-paced reinforcement learning (SPRL), borrowing from the concept of self-paced learning that has been established in the supervised learning literature (Kumar et al., 2010; Jiang et al., 2015; Meng et al., 2017). Klink et al. showed a connection between a regularized RL objective and a sequence of task distributions that trade-off between yielding high expected reward and tasks likely under the target distribution. This interpolant has, however, so far been restricted to Gaussian distributions (Klink et al., 2020a;b; 2021). While successful in experimental evaluations, this Gaussian assumption clearly imposes a limitation on the flexibility of the curriculum and disconnects the algorithmic implementation from the established theory. This disconnect raises the question whether the observed performance of SPRL is due to the Gaussian approximation.

**Contribution:** The key insight presented in this paper is that the Gaussian approximation of existing SPRL implementations is indeed important for their empirical performance. We show that

- Parametric assumptions in SPRL hinder the learning performance in task spaces with non-Gaussian target distributions.
- SPRL can, however, fail to facilitate learning on the target task distributions when leaving these parametric assumptions behind.
- Equipping SPRL with Wasserstein metrics allows for a flexible, particle-based representation of the task distribution that ensures a meaningful interpolation in aforementioned failure cases, providing higher performance.

## 2 RELATED WORK

The main focus of this work is on self-paced reinforcement learning (SPRL, Klink et al. (2020a;b; 2021)) that takes the concept of self-paced curriculum learning (Kumar et al., 2010) from supervised-to reinforcement learning (RL). Opposed to supervised learning, where there is ongoing discussion about the mechanics of curricula and their effect in different situations (Weinshall & Amir, 2020; Wu et al., 2021), the mechanics seem to be more agreed upon in RL. In RL, curricula improve learning performance of an agent by adapting the training environments to its proficiency, and with that e.g. bypass poor exploratory behaviour of non-proficient agents. Applications are by now widely spread and different terms have been established. Adaptive Domain Randomization (Akkaya et al., 2019) uses curricula to gradually diversify training parameters of a simulator to facilitate sim-to-real transfer. Unsupervised environment discovery (Dennis et al., 2020; Jiang et al., 2021b;a) similarly aims to efficiently train an agent which is robust to variations in the environment. Automatic curriculum learning methods (Florensa et al., 2017; Sukhbaatar et al., 2018; Florensa et al., 2018; Portelas et al., 2019; Zhang et al., 2020; Racaniere et al., 2020; Eimer et al., 2021), to which SPRL belongs to, particularly focus on improving the learning speed and/or performance of an agent on a set of desired tasks. Curricula are often generated as distributions that maximize a certain surrogate objective, such as learning progress (Baranes & Oudeyer, 2010; Portelas et al., 2019), intermediate task difficulty (Florensa et al., 2018), regret (Jiang et al., 2021b) or disagreement between $Q$-functions (Zhang et al., 2020). Curriculum generation can also be interpreted as a two-player game (Sukhbaatar et al., 2018). The work by Jiang et al. (2021a) even hints to a link between surrogate objectives and two-player games. Opposed to these interpretations, SPRL has been shown to perform an interpolation between task distributions by Klink et al. (2021), allowing to formally relate the effect of SPRL to the concept of annealing in statistics (Neal, 2001) and homotopic continuation methods in optimization (Allgower & Georg, 2003). We wish to add to this formal understanding of SPRL by investigating the interpolation that it produces more closely.

As this investigation will lead us to the problem of optimal transport, we wish to point out important literature in this field. Dating back to the work by Monge in the 18th century, optimal transport has been understood as an important fundamental concept touching upon many fields in both theory and application (Liu et al., 2019; Peyré et al., 2019; Chen et al., 2021). In probability theory, optimal transport translates to the so-called Wasserstein metric (Kantorovich, 1942) that compares two distributions under a given metric on the sample space. From a computational perspective, entropy-regularized Wasserstein metrics (Cuturi, 2013) have led to tangible speed-ups in computations revolving around optimal transport and are hence widely applied (Feydy et al., 2019).

## 3 PRELIMINARIES

This section serves to introduce the necessary background on (contextual) RL, self-paced RL and optimal transport.

### 3.1 CONTEXTUAL REINFORCEMENT LEARNING

Contextual reinforcement learning (Hallak et al., 2015) can be seen as a conceptual extension to the (single task) reinforcement learning (RL) problem

$$\max_{\pi} J(\pi) = \max_{\pi} \mathbb{E}_{p_0(\mathbf{s}_0), p(\mathbf{s}_{t+1}|\mathbf{s}_t, \mathbf{a}_t), \pi(\mathbf{a}_t|\mathbf{s}_t)} \left[ \sum_{t=0}^{\infty} \gamma^t r(\mathbf{s}_t, \mathbf{a}_t) \right], \tag{1}$$

which aims to maximize the above expected reward objective by finding an optimal a policy $\pi : \mathcal{S} \times \mathcal{A} \mapsto \mathbb{R}$ for a given MDP $\mathcal{M} = \langle \mathcal{S}, \mathcal{A}, p, r, p_0 \rangle$ with initial state distribution $p_0$ and transition dynamics $p$. Contextual RL extends this objective to a space of MDPs $\mathcal{M}(\mathbf{c}) = \langle \mathcal{S}, \mathcal{A}, p_{\mathbf{c}}, r_{\mathbf{c}}, p_{0,\mathbf{c}} \rangle$ equipped with a distribution $\mu{:}\mathcal{C} \mapsto \mathbb{R}$ over contextual variables $\mathbf{c} \in \mathcal{C}$

$$\max_{\pi} J(\pi, \mu) = \max_{\pi} \mathbb{E}_{\mu(\mathbf{c})} \left[ J(\pi, \mathbf{c}) \right]. \tag{2}$$

The policy $\pi : \mathcal{S} \times \mathcal{C} \times \mathcal{A} \mapsto \mathbb{R}$ is conditioned on the contextual parameter $\mathbf{c}$. The distribution $\mu(\mathbf{c})$ encodes the tasks $\mathcal{M}(\mathbf{c})$ that the agent is expected to encounter. Objective $J(\pi, \mathbf{c})$ in Eq. (2) corresponds to the objective $J(\pi)$ in Eq. (1) where, however, the initial state distribution $p_0$, the transition dynamics $p$ as well as the reward function $r$ of $\mathcal{M}$ are replaced by their counterparts in $\mathcal{M}(\mathbf{c})$. This contextual model of optimal decision making is well-suited for learning in multiple related tasks as is the case in multi-task (Wilson et al., 2007), goal-conditioned (Schaul et al., 2015) or curriculum RL (Narvekar et al., 2020).

## 3.2 Self-Paced Reinforcement Learning

Self-paced reinforcement learning (SPRL) has been introduced by Klink et al. (2020a;b; 2021) as a curriculum RL algorithm that alters the context distribution $\mu(\mathbf{c})$ in the contextual RL objective (2) to increase the learning performance of an agent and/or make it less susceptible to local optima of the objective function. SPRL computes a surrogate distribution $p : \mathcal{C} \mapsto \mathbb{R}$ under which to train the RL agent, i.e. optimize $J(\pi, p)$.

This surrogate distribution is found by optimizing the KL divergence to the target distribution $\mu(\mathbf{c})$ subject to two constraints (see Klink et al. (2021, Section 8))

$$\min_{p} \; D_{\mathrm{KL}} \left( p(\mathbf{c}) \,\|\, \mu(\mathbf{c}) \right) \;\; \text{s.t.} \;\; J(\pi, p) \geq \delta \qquad D_{\mathrm{KL}} \left( p(\mathbf{c}) \,\|\, q(\mathbf{c}) \right) \leq \epsilon. \tag{3}$$

The distribution $p(\mathbf{c})$ balances between tasks likely under the (target) distribution $\mu(\mathbf{c})$ and tasks in which the agent currently obtains large rewards. The KL divergence constraint w.r.t. the previous context distribution $q(\mathbf{c})$ prevents large changes in $p(\mathbf{c})$ during subsequent iterations, making the curriculum robust against errors in the estimates of the expected agent performance $J(\pi, \mathbf{c})$.

A particularly interesting aspect of this work is that objective (3) can be interpreted to perform a specific interpolation between the distributions $\mu(\mathbf{c})$, $q(\mathbf{c})$ and a maximum entropy distribution $p_J(\mathbf{c}) \propto \exp(\eta J(\pi, \mathbf{c}))$ encoding high reward tasks. This interpolation is given by

$$p_{\alpha,\eta}(\mathbf{c}) \propto \mu(\mathbf{c})^{\frac{1}{1+\alpha}} q(\mathbf{c})^{\frac{\alpha}{1+\alpha}} \exp(\eta J(\pi, \mathbf{c}))^{\frac{1}{1+\alpha}}. \tag{4}$$

The two parameters $\alpha$ and $\eta$ controlling the interpolation are the Lagrangian multipliers of the two constraints in objective (3). So far, Klink et al. (2020a;b; 2021) restricted the distribution $p_{\alpha,\eta}(\mathbf{c})$ to a Gaussian distributions $p_{\boldsymbol{\nu}}(\mathbf{c}) = \mathcal{N}(\mathbf{c}|\boldsymbol{\mu}, \boldsymbol{\Sigma})$. In this case, optimizing (3) w.r.t. $\boldsymbol{\mu}$ and $\boldsymbol{\Sigma}$ of $p_{\boldsymbol{\nu}}$ corresponds to performing an I-projection of the analytic optimal distribution (4) to the Gaussian restriction, i.e. minimizing $D_{\mathrm{KL}} \left( p_{\boldsymbol{\nu}}(\mathbf{c}) \,\|\, p_{\alpha,\eta}(\mathbf{c}) \right)$ w.r.t. $\boldsymbol{\nu}$.

In this work, we are interested in investigating the distribution $p_{\alpha,\eta}$ outside of this parametric restriction $p_{\boldsymbol{\nu}}$, i.e. truly employing the distribution (4) instead of its I-projection to a Gaussian.

## 3.3 Optimal Transport

The problem of optimally transporting density between two distributions has been initially investigated by Monge (1781). As of today, generalizations established by Kantorovich (1942) have led to so called **Wasserstein distances** as metrics between probability distributions defined on a metric space $M = (d, \mathcal{C})$ with metric $d : \mathcal{C} \times \mathcal{C} \mapsto \mathbb{R}_{\geq 0}$

$$\mathcal{W}_p(p_1, p_2) = \left( \inf_{\gamma \in \Gamma(p_1, p_2)} \int_{\mathcal{C} \times \mathcal{C}} d(\mathbf{c}_1, \mathbf{c}_2)^p \, \mathrm{d}\gamma(\mathbf{c}_1, \mathbf{c}_2) \right)^{1/p}, \qquad p \geq 1$$

$$\Gamma(p_1, p_2) = \left\{ \gamma : \mathcal{C} \times \mathcal{C} \mapsto \mathbb{R}_{\geq 0} \middle| p_1(\mathbf{c}_1) = \int \gamma(\mathbf{c}_1, \mathbf{c}_2) \, \mathrm{d}\mathbf{c}_2, \; p_2(\mathbf{c}_2) = \int \gamma(\mathbf{c}_1, \mathbf{c}_2) \, \mathrm{d}\mathbf{c}_1 \right\}$$

The distance between $p_1$ and $p_2$ results from solving an optimization problem that finds a so-called plan $\gamma$. This plan encodes how to equalize $p_1$ and $p_2$ taking into account the cost of moving density between between parts of the space $\mathcal{C}$. This cost is encoded by the metric $d$. In the following, we

---

**Algorithm 1** Self-Paced Reinforcement Learning Implementations

---

**Input:** Context dist. $p_0(\mathbf{c})$, target context dist. $\mu(\mathbf{c})$, performance threshold $\delta$, distance bound $\epsilon$
**for** $k = 0$ **to** $K$ **do**
    **Agent Improvement:**
    Sample contexts $\mathbf{c}_i \sim p_k(\mathbf{c})$, $i \in [1, M]$
    Train policy $\pi$ under $\mathbf{c}_i$ and observe episodic rewards $R_i = \sum_{t=1}^{\infty} r_{\mathbf{c}_i}(\mathbf{s}_t, \mathbf{a}_t)$, $i \in [1, M]$
    Estimate $J(\pi, \mathbf{c})$ from the dataset $\mathcal{D} = \{(\mathbf{c}_i, R_i) | i \in [1, M]\}$
    **Context Distribution Update:**
    G-SPRL Optimize (3) w.r.t. $\boldsymbol{\mu}_{k+1}$ and $\boldsymbol{\Sigma}_{k+1}$, where $p_{k+1}(\mathbf{c}) = \mathcal{N}(\mathbf{c} | \boldsymbol{\mu}_{k+1}, \boldsymbol{\Sigma}_{k+1})$
    NP-SPRL Optimize (3) w.r.t $\alpha$ and $\eta$ using a discrete approximation $\bar{\mathbf{p}}_{\alpha,\eta} \approx p_{\alpha,\eta}(\mathbf{c})$ (4)
    WB-SPRL Optimize (7) w.r.t. $\boldsymbol{\beta}$ to obtain $p_{\boldsymbol{\beta}}(\mathbf{c})$
**end for**

---

will always assume to work with 2-Wasserstein distances under euclidean metric, i.e. $p = 2$ and $d(\mathbf{c}_1, \mathbf{c}_2) = \|\mathbf{c}_1 - \mathbf{c}_2\|_2$. Particularly interesting for the investigation conducted in this paper are so called **Wasserstein barycenters**

$$p_{\mathcal{W}_2}(\mathbf{c}) = \arg\min_p \sum_{k=1}^{K} w_k \mathcal{W}_2(p, p_k), \qquad \sum_{k=1}^{K} w_k = 1, \tag{5}$$

that perform a weighted interpolation between a set of distributions $p_k$ by computing a distribution $p$ that minimizes the above sum of Wasserstein distances. As illustrated in appendix A, these barycenters allow to smoothly interpolate between complex distributions w.r.t. the defined metric $d$.

# 4 NON-PARAMETRIC SELF-PACED REINFORCEMENT LEARNING

To highlight the importance of a non-parametric variant of SPRL that, nonetheless, respects the metric structure of the context space $\mathcal{C}$, we investigate three versions of SPRL: G-SPRL– the Gaussian version from Klink et al. (2021), NP-SPRL– a faithful implementation of the SPRL objective (3) that computes and uses the distribution $p_{\alpha,\eta}(\mathbf{c})$ (4) by discretizing the context space $\mathcal{C}$, and WB-SPRL– an instantiation of Wasserstein barycenters in the SPRL framework. As shown in Algorithm 1, all implementations only differ in the computation of the context distributions.

## 4.1 NP-SPRL

Computing the analytic distribution $p_{\alpha,\eta}(\mathbf{c})$ and adjusting the parameters $\alpha$ and $\eta$ via the SPRL objective (3) will require approximations in the general case, as sampling arbitrary continuous distributions is an open research problem (Brooks et al., 2011; Liu & Wang, 2016; Liu et al., 2019; Wibisono, 2018). Using approximate methods resulting from this research would, however, directly interfere with our intent of evaluating the behavior of $p_{\alpha,\eta}(\mathbf{c})$ as exactly as possible. Consequently, we discretize the continuous context spaces in our experiments in order to faithfully sample and evaluate expectations w.r.t. $p_{\alpha,\eta}(\mathbf{c})$. This discretization is schematically shown in Figure 1 for one of the evaluation environments. Although such a discretization will clearly not scale gracefully to higher dimensions, it allows us to investigate the "analytic" behavior of SPRL, which, as we will show, can be sub-optimal even for low-dimensional context spaces.
With a context space $\mathcal{C} \subseteq \mathbb{R}^d$ discretized into $N$ cells, we can represent $p_{\alpha,\eta}$ as a vector $\bar{\mathbf{p}}_{\alpha,\eta} \in \mathbb{R}_{\geq 0}^N$. To sample a continuous context $\mathbf{c} \in \mathcal{C}$, we can then first sample an index of a cell from $\bar{\mathbf{p}}_{\alpha,\eta}$ and then sample uniformly within this cell to obtain a context $\mathbf{c}$. Further, the KL divergences in the SPRL objective (3) are straightforward to evaluate when working with $\bar{\mathbf{p}}_{\alpha,\eta}$. To evaluate the expected performance, we simply evaluate the performance measure $J(\pi, \mathbf{c})$ at the $N$ cell centers $\bar{\mathbf{c}}_n$ to obtain the vector $\bar{\mathbf{J}}(\pi) \in \mathbb{R}^N$. With that we obtain $J(\pi, p_{\alpha,\eta}) = \bar{\mathbf{p}}_{\alpha,\eta}^T \bar{\mathbf{J}}(\pi)$.

## 4.2 WB-SPRL

To leverage optimal transport in an SPRL style algorithm, we realize the interpolation between the current distribution $q(\mathbf{c})$, target distribution $\mu(\mathbf{c})$ and "value distribution" $p_J(\mathbf{c}) \propto \exp(\eta J(\pi, \mathbf{c}))$ by Wasserstein barycenters

$$p_{\boldsymbol{\beta}}(\mathbf{c}) = \arg\min_p (1 - \beta_1 - \beta_2) \mathcal{W}(p(\mathbf{c}), \mu(\mathbf{c})) + \beta_1 \mathcal{W}(p(\mathbf{c}), p_J(\mathbf{c})) + \beta_2 \mathcal{W}(p(\mathbf{c}), q(\mathbf{c})). \tag{6}$$

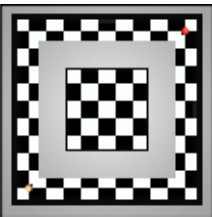 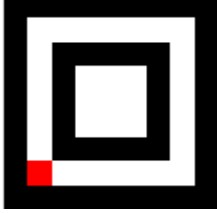 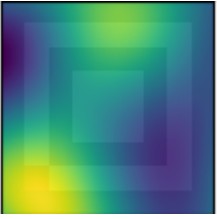 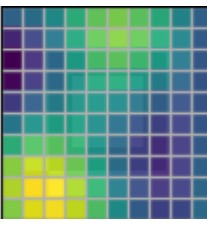

(a) Maze Environment      (b) Context Space $\mathcal{C}$      (c) Function $f$ on $\mathcal{C}$      (d) $f$ discretized on $\mathcal{C}$

Figure 1: (a) The first environment for evaluation of SPRL is a maze task simulated in MuJoCo (Todorov et al., 2012), in which a point-mass needs to move in a maze of circular shape to reach desired target positions. (b) The target position is encoded via the contextual variable $\mathbf{c} \in \mathcal{C} \subseteq \mathbb{R}^2$. The area highlighted in red visualizes the initial position of the point mass and the walls of the environment are shown in black. (c + d) In order to faithfully evaluate expectations over functions or compute KL divergences e.g. in the SPRL objective (3), we discretize the context space $\mathcal{C}$.

The weights $\boldsymbol{\beta}$ of the interpolation are adjusted with the goal of minimizing $\mathcal{W}_2(p_{\boldsymbol{\beta}}, \mu)$ while ensuring a constraint on expected performance and distance to the previous distribution $q(\mathbf{c})$

$$\min_{\boldsymbol{\beta}} \ \mathcal{W}_2(p_{\boldsymbol{\beta}}(\mathbf{c}), \mu(\mathbf{c})) \ \text{ s.t } \ J(\pi, p_{\boldsymbol{\beta}}) \geq \delta \quad \mathcal{W}_2(p_{\boldsymbol{\beta}}(\mathbf{c}), q(\mathbf{c})) \leq \epsilon. \tag{7}$$

A difference w.r.t. SPRL is that the parameter $\eta$ of the value distribution $p_J(\mathbf{c})$ is not adjusted in optimization problem (7). Instead we adjust $\eta$ before optimizing (7) such that $J(\pi, p_J) \geq \delta_H$, where $\delta_H > \delta$ is another performance threshold. This ensures that with $\beta_1 \to 1$, it holds that $J(\pi, p_{\boldsymbol{\beta}}) \geq \delta$. We resort to particle-based representations of the distributions when implementing (6) and (7). This allows us to make use of Monge maps to compute $p_{\boldsymbol{\beta}}(\mathbf{c})$ efficiently. More details on the implementation of WB-SPRL are provided in appendix C.

## 5   EXPERIMENTS

The experiments in this section serve to show the need for a realization of SPRL without parametric restrictions on $p_{\alpha,\eta}$ but to also highlight that NP-SPRL is not necessarily well-suited for this endeavour due its ignorance w.r.t. the metric on $\mathcal{C}$. Experimental details can be found in appendix D. *Code is provided in the supplementary material and will be made available upon acceptance.* To situate our method in the field of curriculum RL and to showcase how our method performs w.r.t. the current state-of-the art, we further evaluate ACL, GOALGAN, ALP-GMM, VDS and PLR (Graves et al., 2017; Florensa et al., 2018; Portelas et al., 2019; Zhang et al., 2020; Jiang et al., 2021b) alongside the investigated SPRL variants.

### 5.1   MAZE

We first turn to a sparse-reward, maze-like environment depicted in Figure 1, in which an agent needs to reach a desired goal. Such environments have e.g. been investigated by Florensa et al. (2018). The contexts $\mathbf{c} \in \mathcal{C}$ of this environment encode the goal position to be reached and hence contains unsolvable contexts (goals inside of a wall or in the inner circle of the maze). Defining $\mu(\mathbf{c})$ to be uniform over the context space, the curriculum needs to identify and train the agent on the subspace of feasible tasks in order to achieve a good learning performance. This subspace of feasible tasks is highly non-Gaussian, making it an interesting testbed for NP- and WB-SPRL. We discretize the context space $\mathcal{C} \subseteq \mathbb{R}^2$ of goal positions into a $50 \times 50$ grid. Figure 2 compares the performance of the different CRL algorithms. We see that both NP- and WB-SPRL perform better than G-SPRL, since the performance constraint $\mathbb{E}_{p(\mathbf{c})}[J(\pi, \mathbf{c})] \geq \delta$ in objective (3) at some point prevents the Gaussian context distribution of G-SPRL from expanding, as otherwise too many infeasible tasks would be encoded in $p_{\boldsymbol{\nu}}(\mathbf{c})$ and hence the performance constraint violated. As shown in Figures 2 and 7, the non-parametric versions of SPRL can flexibly assign probability to feasible contexts, resulting in high learning performance.

### 5.2   POINT-MASS

We now consider the point-mass environment investigated by (Klink et al., 2020a;b; 2021). As shown in Figure 3a, a point mass needs to be steered through a narrow gate to reach a goal position

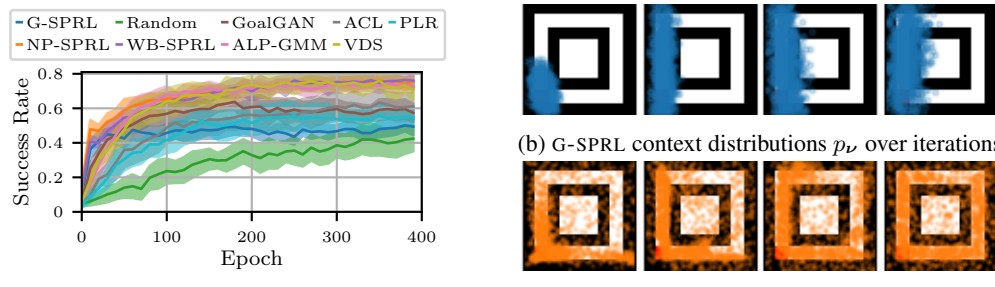

(a) Learning Performance (Maze)

(b) G-SPRL context distributions $p_{\boldsymbol{\nu}}$ over iterations

(c) NP-SPRL context distributions $p_{\alpha,\eta}$ over iterations

Figure 2: a) Achieved success rate of different curricula and a uniform sampling baseline (Random) over iterations. Mean and standard error are computed from 10 runs. b) Parametric and non-parametric context distributions $p_{\boldsymbol{\nu}}(\mathbf{c})$ and $p_{\alpha,\eta}(\mathbf{c})$ for a run of G-SPRL and NP-SPRL respectively. The distributions are represented by 2000 samples drawn from them.

on the other side of a wall. While Klink et al. only considered a narrow gate at one specific position as the target task, we will investigate a version in which the gate is located at one of two opposing positions $\mathbf{c}_1 = [-3\ 0.5]$ and $\mathbf{c}_2 = [3\ 0.5]$, making $\mu(\mathbf{c})$ a bi-modal distribution. This again challenges the Gaussian restriction of $p_{\boldsymbol{\nu}}(\mathbf{c})$ in the G-SPRL algorithm. We again discretize the context space $\mathcal{C} \subseteq \mathbb{R}^2$ into a grid of 50 bins on each axis. We investigate two target distributions with different log-likelihoods, that nonetheless produce very similar samples

$$\mu_1(\mathbf{c}) = \frac{1}{2}\mathcal{N}\left(\mathbf{c}_1, 10^{-4}\mathbf{I}\right) + \frac{1}{2}\mathcal{N}\left(\mathbf{c}_2, 10^{-4}\mathbf{I}\right) \qquad \mu_2(\mathbf{c}) = \begin{cases} \frac{1}{2} & \text{, if } \mathbf{c} \in \{\mathbf{c}_1, \mathbf{c}_2\} \\ \approx 0 & \text{, else.} \end{cases}$$

Note that "$\approx 0$" corresponds to a value of $\exp(-1000)$, as this ensures that $\mu(\mathbf{c})$ is absolutely continuous w.r.t. $p_{\alpha,\eta}(\mathbf{c})$ and $p_{\boldsymbol{\nu}}(\mathbf{c})$, which is required to compute the KL divergence between them. Figure 3 shows that the performance of G- and NP-SPRL depends drastically on the choice of target distribution. Further, we see that NP-SPRL does not outperform G-SPRL in this environment, although it should be able to match the bi-modal target distributions, which G-SPRL cannot. Finally, we see that WB-SPRL outperforms both other versions of SPRL and achieves a consistent performance across target distributions. The illustrations in Figures 4 and 5 serve to illustrate the underlying problem. As evident in Figure 4, NP-SPRL is indeed able to match the target distribution $\mu_1(\mathbf{c})$ correctly while G-SPRL ultimately covers only one mode of the target density $\mu_1(\mathbf{c})$. As shown in appendix B, NP-SPRL still only learns to solve one of the two tasks likely under $\mu_1(\mathbf{c})$. Investigating Figure 4b more closely reveals that NP-SPRL only generates a "proper" curriculum for one of the two target tasks (the left one in the images) by gradually interpolating between easy tasks and the target task, while simply incorporating the second target task into $p_{\alpha,\eta}$ **without** generating an appropriate curriculum for the agent to learn this task. Figure 5 shows that the curricula generated by G- and NP-SPRL for $\mu_2(\mathbf{c})$ are even more problematic. G-SPRL increases the variance of the Gaussian context distribution to match the constant (negligible) likelihood that is assigned to the non-target

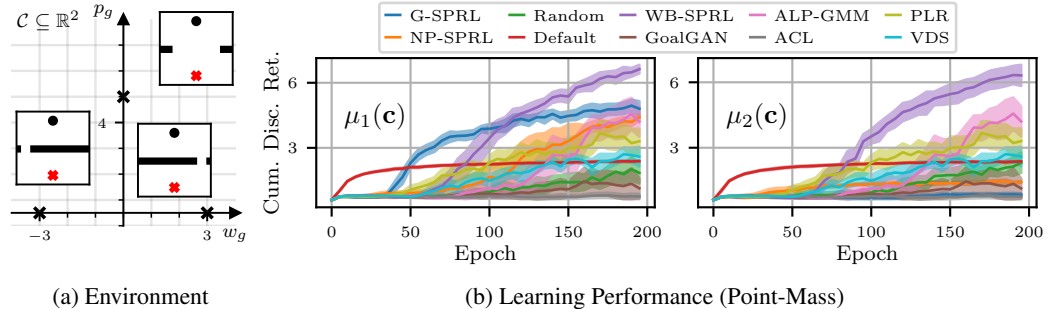

(a) Environment

(b) Learning Performance (Point-Mass)

Figure 3: (a) The point mass environment with its two-dimensional context space. The target distributions $\mu_1(\mathbf{c})$ and $\mu_2(\mathbf{c})$ encode the two gates with width $w_g = 0.5$, in which the agent (black dot) is required to navigate through a narrow gate at different positions to reach the goal (red cross). (b) Discounted cumulative return over iterations obtained under different curricula for both distributions. Statistics (mean and standard error) are computed from 10 seeds.

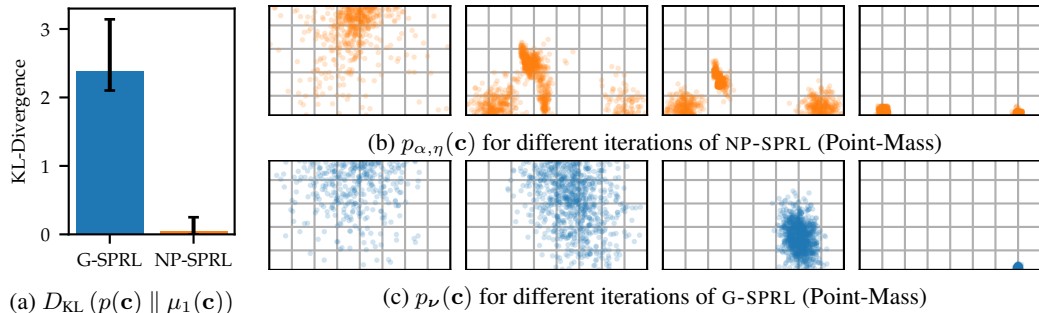

(a) $D_{\text{KL}}\left(p(\mathbf{c}) \| \mu_1(\mathbf{c})\right)$       (b) $p_{\alpha,\eta}(\mathbf{c})$ for different iterations of NP-SPRL (Point-Mass)

(c) $p_{\boldsymbol{\nu}}(\mathbf{c})$ for different iterations of G-SPRL (Point-Mass)

Figure 4: (a) Mean KL divergences between $\mu_1(\mathbf{c})$ and the final context distribution computed by G- and NP-SPRL as well as the minimum and maximum over 10 seeds. (b) and (c) visualize $p_{\alpha,\eta}(\mathbf{c})$ of NP-SPRL and $p_{\boldsymbol{\nu}}(\mathbf{c})$ of G-SPRL for increasing iterations of the algorithms (left to right).

tasks, while NP-SPRL now interpolates between none of the target tasks. Consequently, neither the agent under G- nor NP-SPRL learns to solve any of the target tasks. The observed shortcomings of G- and NP-SPRL are caused by their use of the KL divergence to measure similarity between distributions. Figure 6 shows that the KL divergence generates interpolations that are highly dependent on the likelihood function of the distributions, as the KL divergence does not take the metric structure of the sample space $\mathcal{C}$ into account. Indeed Gaussian distributions encode a (Euclidean) metric via their log-likelihood, explaining the acceptable performance of G- and NP-SPRL for $\mu_1(\mathbf{c})$. Looking at Figure 6b and the curricula generated by WB-SPRL in Figure 7, we see that the explicit notion of a (Euclidean) metric in WB-SPRL leads to a gradual change in the tasks encoded by the curriculum for both target distributions. This leads to higher learning performance as visualized in Figure 3.

## 5.3 PICK AND PLACE

Next, we consider the pick-and-place task in the OpenAI gym environment suite (Brockman et al., 2016) in which a robot is tasked to grasp a block on a table and move it to a desired position (see Figure 8). The sparse reward of this environment, only rewarding the robot upon completing the desired task, makes it a very challenging exploration problem, as can be seen in Figure 8 in which (default) SAC does not learn this pick and place task. One way to alleviate such challenging exploration problems is to learn the task via a curriculum of starting states. Such starting states can be obtained from an expert demonstration, i.e. a trajectory $\boldsymbol{\tau}_{\text{EXPERT}} : [0,1] \mapsto \mathcal{S}$. A curriculum over this trajectory is then formally defined by choosing the context space to be the unit interval $\mathcal{C} = [0,1] \subseteq \mathbb{R}$. The contextual parameter $c$ only influences the initial state distribution $p_{0,c} = \delta_{\mathbf{s}=\boldsymbol{\tau}_{\text{EXPERT}}(c)}$. We investigate the NP- and WB-SPRL algorithms in this setting by recording one execution of a hand-crafted controller that first moves the end-effector above the block to be grasped, then lowers the end-effector, grasps the block and moves to the target. We slightly randomize the position of the goal as well as the block to enforce that the agent learns a robust policy. In a realistic scenario, we only record the expert demonstration at a discrete set of states $\{t_i | i = 1, \dots, N\}$ and hence we have a truly discrete CRL setting in this experiment very well suited for NP-SPRL. Furthermore, the finite number of

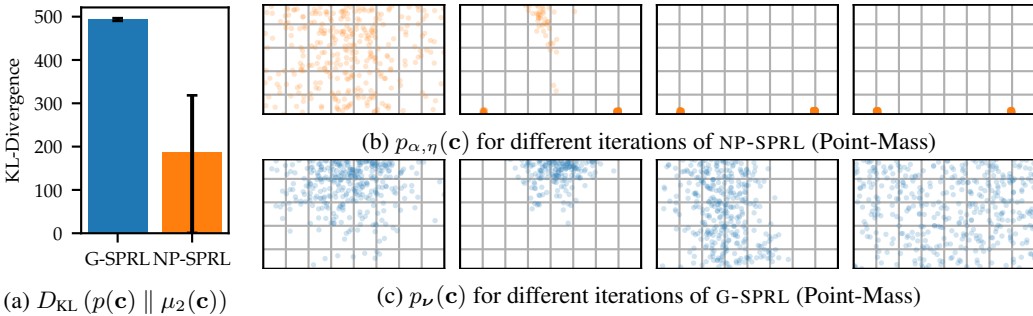

(a) $D_{\text{KL}}\left(p(\mathbf{c}) \| \mu_2(\mathbf{c})\right)$       (b) $p_{\alpha,\eta}(\mathbf{c})$ for different iterations of NP-SPRL (Point-Mass)

(c) $p_{\boldsymbol{\nu}}(\mathbf{c})$ for different iterations of G-SPRL (Point-Mass)

Figure 5: (a) Mean KL divergences between $\mu_2(\mathbf{c})$ and the final context distribution computed by G- and NP-SPRL as well as the minimum and maximum over 10 seeds. (b) and (c) visualize $p_{\alpha,\eta}(\mathbf{c})$ of NP-SPRL and $p_{\boldsymbol{\nu}}(\mathbf{c})$ of G-SPRL for increasing iterations of the algorithms (left to right).

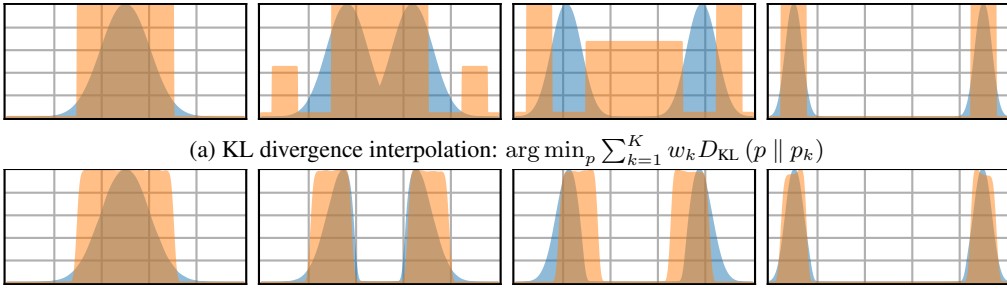

(a) KL divergence interpolation: $\arg\min_p \sum_{k=1}^{K} w_k D_{\mathrm{KL}}(p \parallel p_k)$

(b) Wasserstein barycenter (Equation 5)

Figure 6: Interpolations between unimodal (left) and bi-model (right) distributions $p_1(\mathbf{c})$ and $p_2(\mathbf{c})$ via a KL divergence-based interpolation and Wasserstein barycenters (5). In one case, all distribution are Gaussians or mixture of Gaussians (blue). In the other case, the distributions are uniform distributions (orange). The Wasserstein barycenters are computed using a particle-based approximation (Feydy et al., 2019). The visualized PDFs are then estimated using a kernel density estimation. This results in a small amount of smoothing for the uniform distribtions.

contexts avoids the necessity for function approximators to approximate the expected performance $J(\pi, \mathbf{c})$ in a context $\mathbf{c}$, as we can simply estimate the performance in each (discrete) context via a sliding window. We again investigate two target context distributions, one being a narrow Gaussian target distribution with mean at $c = 0$ and negligible variance ($\mu_1(\mathbf{c})$) and the other being again a Dirac-delta with a negligible amount of probability in any context to be absolutely continuous w.r.t. any distribution ($\mu_2(\mathbf{c})$). The results in Figure 8 show that WB-SPRL generates a curriculum that allows the agent to learn a reliable policy by smoothly moving probability from later steps of the expert demonstration towards earlier ones. This learned policy is more reliable ($25\%$ vs. $100\%$ success rate) and faster ( 69 vs. 13 steps for task completion) than the demonstration. Both default learning (always starting from the initial state) and NP-SPRL do not lead to successful policies. In appendix B, we show that the non-effectiveness of NP-SPRL is again grounded in the degenerate interpolation that puts no probability mass at intermediate time-steps regardless of the target distribution.

## 6 TEACHMYAGENT BENCHMARK

A final environment for evaluation is the TeachMyAgent benchmark proposed by (Romac et al., 2021). In their evaluations accompanying the benchmark, Romac et al. reported a poor performance of G-SPRL compared to other automatic curriculum RL methods. Consequently, we want to investigate the improvement of WB-SPRL in this benchmark to verify the observed benefits and identify further directions for improving the SPRL framework. Figure 9 shows that WB-SPRL significantly improves upon G-SPRL, achieving top performance across the methods in the *mostly unfeasible* scenario. We see that WB-SPRL particularly lacks behind ALP-GMM in the *mostly trivial* setting. Figure 9c shows that WB-SPRL quickly reduces to uniform sampling in this setting (on average after 60 out of 370 iterations), as it can incorporate all tasks while fulfilling the desired performance threshold. This observation explains the similar performance between random sampling and WB-SPRL in the *mostly trivial* setting and further highlights an important conceptual difference between SPRL and

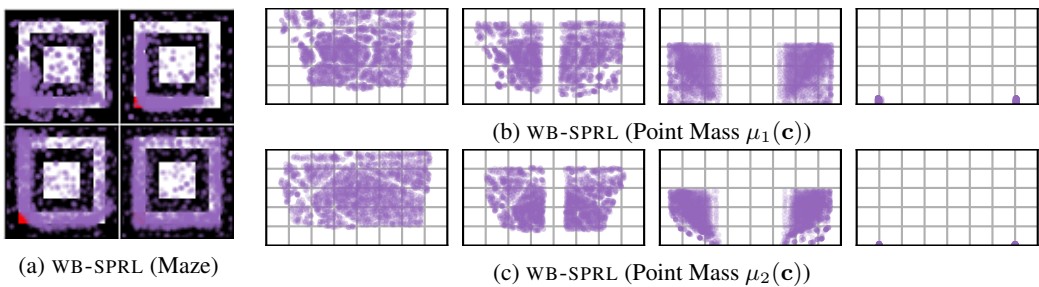

(a) WB-SPRL (Maze)

(b) WB-SPRL (Point Mass $\mu_1(\mathbf{c})$)

(c) WB-SPRL (Point Mass $\mu_2(\mathbf{c})$)

Figure 7: Visualizations of the empirical distributions $p_\beta$ generated by WB-SPRL in the maze- (a) and point-mass environments for both target distributions $\mu_1(\mathbf{c})$ and $\mu_2(\mathbf{c})$ ((b) + (c)).

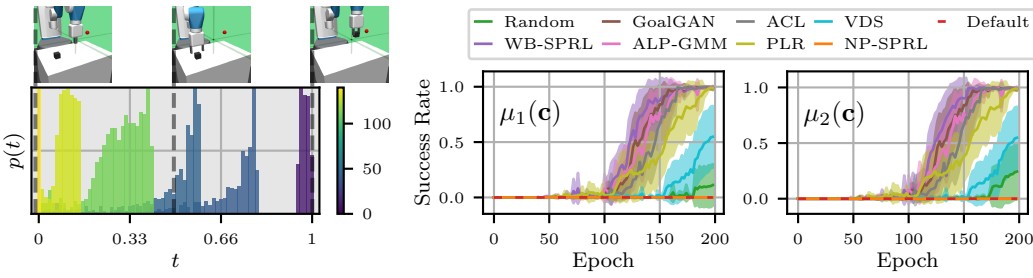

(a) WB-SPRL Curriculum (Pick and Place)  (b) Learning Performance (Pick and Place)

Figure 8: (a) A WB-SPRL curriculum for the pick-and-place task. The vertical dotted lines and small images visualize the states corresponding to different time steps of the expert trajectory. The colored bars indicate the context distributions $p_\beta(\mathbf{c})$ for different iterations of WB-SPRL. The correspondence between color and algorithm is shown via the colorbar on the right. (b) Success rates over iterations for different learning algorithms in the pick and place task for target distribution $\mu_1(\mathbf{c})$ and $\mu_2(\mathbf{c})$. Mean and standard error is computed from 10 algorithm runs.

ALP-GMM: The maximization of learning progress in ALP-GMM leads to a prioritized sampling of the target (uniform distribution), efficiently avoiding tasks the agent has already learned. SPRL does not have a notion of avoiding "too easy" tasks as of now. Introducing such a prioritized sampling of $p(\mathbf{c})$ is hence an interesting future research direction and may allow to combine the benefits of the SPRL framework with other successful curriculum RL methods.

# 7 CONCLUSION

We investigated self-paced reinforcement learning (SPRL) outside of its current parametric restrictions, revealing important shortcomings of the KL divergence for measuring the similarity of (task) distributions in curriculum reinforcement learning. We showed that replacing the KL divergences, originally employed in SPRL, with Wasserstein distances alleviates the observed problems of SPRL. Apart from achieving higher empirical performance in the SPRL framework, our results indicate that introducing a notion of metric structure on the task space may be an important next step for deriving a principled, yet practical, understanding of curriculum RL. Our findings motivate a variety of future investigations, such as more elaborate implementations of WB-SPRL that e.g. directly move the individual particles of the context distribution via the gradient of the performance measure $\nabla_\mathbf{c} J(\pi, \mathbf{c})$ instead of using the value distribution $p_J(\mathbf{c})$ as a proxy. As highlighted in Section 6, importance-sampling of the context distributions are expected to further improve performance. Finally, investigating other metrics than the Euclidean one considered in this paper is expected to allow for principled interpolations in non-Euclidean spaces.

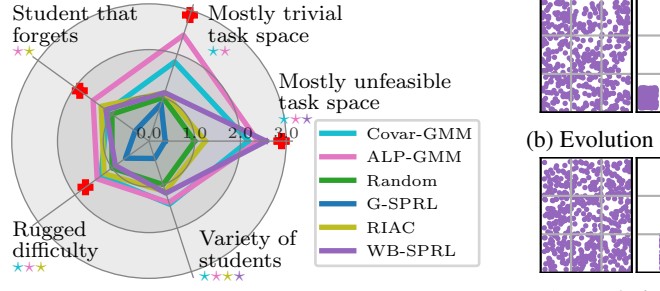

(a) Results in the TeachMyAgent benchmark

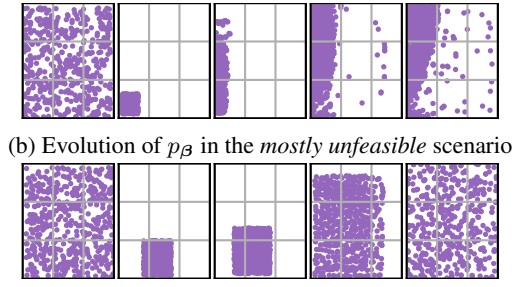

(b) Evolution of $p_\beta$ in the *mostly unfeasible* scenario

(c) Evolution of $p_\beta$ in the *mostly trivial* scenario

Figure 9: a) Performance on the TeachMyAgent benchmark in the *no expert knowledge* setting. Baseline results are taken from (Romac et al., 2021). Please refer to (Romac et al., 2021) for a detailed explanation of the setup. b) + c) Visualization of the WB-SPRL curricula in the *mostly unfeasible* and the *mostly trivial* task space scenario. The $x$-axis encodes the height of the obstacles that the agents encounters while the $y$-axis encodes the distance between them.

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

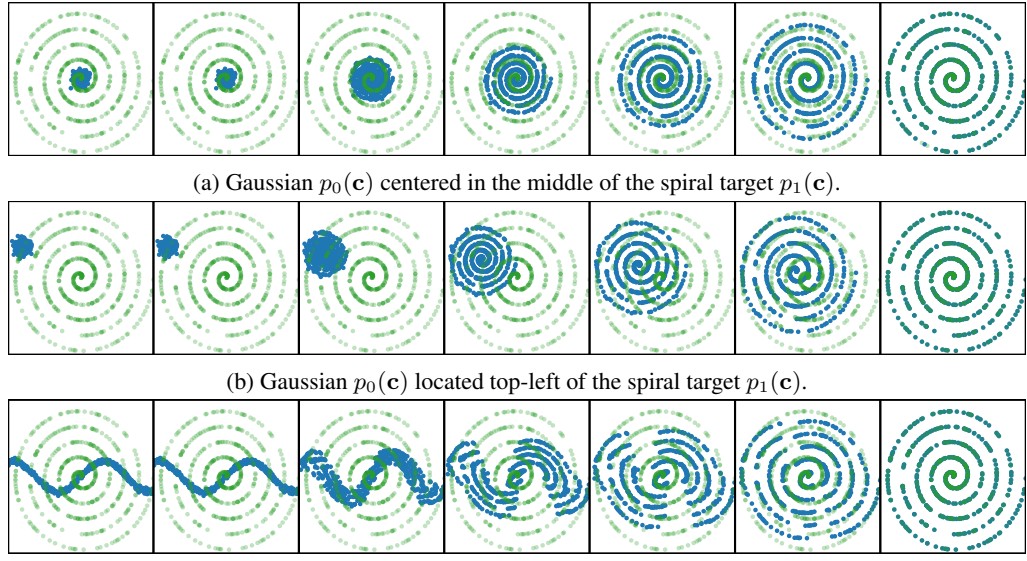

(a) Gaussian $p_0(\mathbf{c})$ centered in the middle of the spiral target $p_1(\mathbf{c})$.

(b) Gaussian $p_0(\mathbf{c})$ located top-left of the spiral target $p_1(\mathbf{c})$.

(c) Sine wave with Gaussian noise $p_0(\mathbf{c})$.

Figure 10: Wasserstein barycenters between two distributions $p_0(\mathbf{c})$ and $p_1(\mathbf{c})$. The above figures visualize the barycenters $p_w(\mathbf{c}) = \arg\min_p w\mathcal{W}_2(p, p_1) + (1 - w)\mathcal{W}_2(p, p_0)$ for $w = 0, 0.2, 0.4, 0.6, 0.8, 1$ (from left to right). The target $p_1(\mathbf{c})$ is shown in green, while the respective barycenter $p_w(\mathbf{c})$ is shown in blue.

## A  ILLUSTRATION OF WASSERSTEIN BARYCENTERS

This short appendix serves to showcase the capacity of Wasserstein barycenters to interpolate between complicated target distributions. Figure 10 visualizes different barycenters that interpolate between different source distributions and a target distribution encoding samples contained in a spiral. Modelling such a spiral, e.g. via Gaussian mixture models would require many components to accurately model the narrow, curved line to which the samples are constrained. As we represent the distributions via a set of particles (see appendix C), the barycenter computation does not require knowledge of the log-likelihood function of the target distribution but only requires samples from it, which is beneficial is the specification of the likelihood function is challenging. We see that the barycenters gradually displace the particles to match the target distribution. For the sinusoidal source distribution in Figure 10c, we can see that the idea of minimum action, which is inherent to Wasserstein metrics, creates non-trivial interpolations that are still smooth w.r.t. the defined metric (in this case the euclidean metric on $\mathbb{R}^2$).

## B  QUALITATIVE RESULTS

Figure 11 shows trajectories that have been generated by agents trained with different curricula. We see that directly learning on the two target tasks prevents the agent from finding the gates in the wall to pass through. Consequently, the agent minimizes the distance to the goal by moving right in front of the wall (but not crashing into it) to accumulate reward over time. We see that random learning indeed generates meaningful behavior. This behavior is, however, not precise enough to pass reliably through the wall. As mentioned in the main paper, G-SPRL and NP-SPRL only learn to pass through one of the gates. We see that in the particular run displayed in Figure 11, NP-SPRL stayed closer to the wall than G-SPRL in those contexts in which it could not pass the wall. This better behavior is probably a result of NP-SPRL ultimately sampling both target tasks during training while G-SPRL only focuses on one. Finally, WB-SPRL learns a policy that can pass through both gates reliably, showing that the gradual interpolation towards both target tasks allowed to learn both of them. ALP-GMM, PLR and VDS also learn good policies. The generated trajectories are, however, not as precise as the ones learned with WB-SPRL. The reason for this lack of precision is probably that these methods have no notion of target distribution and hence cannot focus learning on the target

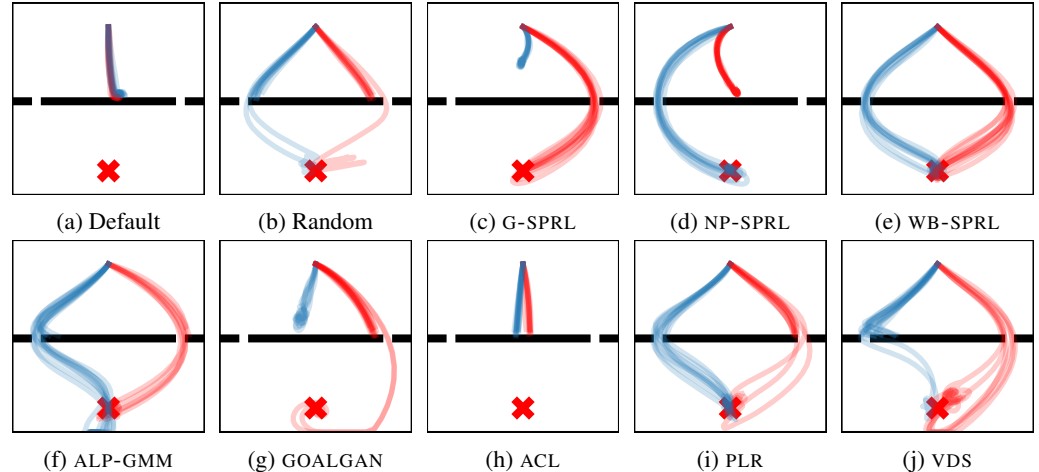

Figure 11: Final trajectories generated by the different investigated curricula in the point mass environment. The color encodes the context: Blue represents gates positioned at the left and red gates positioned at the right.

tasks at later iterations. GOALGAN partly creates meaningful behavior. However, this behavior is unreliable and hence leads to low returns due to the agent frequently crashing into the wall. ACL does not learn to solve the task. The reason for the poor performance of ACL in this task is probably explained by the fact that ACL models the curriculum generation as a bandit problem, which in the $50 \times 50$ grid used in the point mass experiment needs to choose between 2500 arms.

Figure 12 shows the distribution $p_{\alpha,\eta}(\mathbf{c})$ generated by NP-SPRL for different iterations in the pick and place environment. We see that regardless of the target distribution, the interpolation skips all intermediate states of the expert demonstration. This is surprising, given that the interpolation worked at least somewhat desirably in the point mass environment for a Gaussian target distribution. Comparing Figures 12a and 12b, we see that the Gaussian target distribution leads to $p_{\alpha,\eta}(\mathbf{c})$ putting probability mass on a few more states than only the initial one. However, these states are still so close to the initial state that no learning takes place.

## C    IMPLEMENTATION DETAILS OF WB-SPRL

The implementation of WB-SPRL relies on the computation of the Wasserstein barycenter $p_{\boldsymbol{\beta}}(\mathbf{c})$. To compute this barycenter, we represent the distributions $\mu(\mathbf{c})$, $p_J(\mathbf{c})$ and $q(\mathbf{c})$ via a set of particles. For a distribution $p(\mathbf{c})$, we denote the particle-based representation as $\tilde{p}(\mathbf{c}) = \frac{1}{N} \sum_{n=1}^{N} \delta_{\mathbf{c}_n^p}(\mathbf{c})$ where $\mathbf{c}_n^p \sim p(\mathbf{c})$ and $\delta_{\mathbf{c}_n^p}$ is a Dirac delta located at $\mathbf{c}_n^p$. This representation allows to compute the so-called Monge maps $T_{\tilde{p}_1 \to \tilde{p}_k} : \mathcal{C} \mapsto \mathcal{C}$ that encode how to move the particles from the distribution $\tilde{p}_1$ to $\tilde{p}_k$. In the case of the Wasserstein-2 distances, we can use a linear combination of these Monge maps as a reasonable approximation to the exact barycenter $p_{\boldsymbol{\beta}}(\mathbf{c})$

$$\tilde{p}_{\boldsymbol{\beta}}(\mathbf{c}) \approx (\beta_1 T_{\tilde{q} \to \tilde{p}_J} + \beta_2 T_{\tilde{q} \to \tilde{\mu}})_\# \, \tilde{q}(\mathbf{c})$$

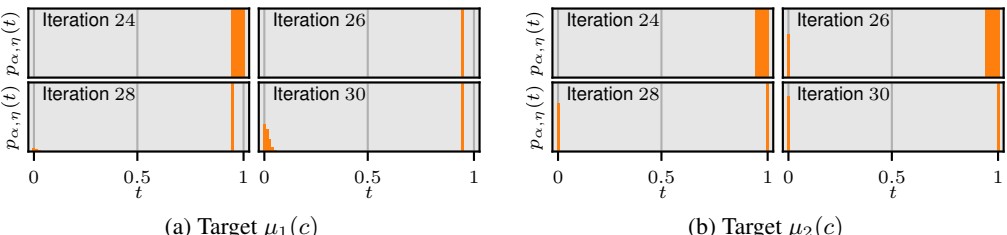

Figure 12: Visualization of the curriculum distribution $p_{\alpha,\eta}(\mathbf{c})$ of NP-SPRL for different iterations in the pick and place tasks. (a) shows the curriculum generated for the Gaussian target distribution $\mu_1(\mathbf{c})$. (b) shows the curriculum in case of using the Dirac-delta-like distribution $\mu_2(\mathbf{c})$.

| | | GENERAL | | | G-SPRL | | NP-SPRL | | WB-SPRL |
|---|---|---|---|---|---|---|---|---|---|
| ENV. | $\delta$ | $\epsilon_{\mathrm{KL}}\|\epsilon_{\mathrm{M}}$ | $n_{\mathrm{STEP}}$ | $n_{\mathrm{OFFSET}}$ | $\boldsymbol{\sigma}_{\mathrm{LB}}$ | $D_{\mathrm{KL}_{\mathrm{LB}}}$ | $\boldsymbol{\sigma}_{\mathrm{LB}}$ | $D_{\mathrm{KL}_{\mathrm{LB}}}$ | $\delta_H$ |
| MAZE | .65 | .25\|1 | 10000 | 5 | - | - | - | - | .9 |
| POINT-MASS | 4 | .25\|.25 | 4096 | 5 | [.2 .1875] | 8000 | 4 | 1000 | 6 |
| PNP | .5 | .5\|.13 | 3000 | 25 | - | - | - | - | .8 |
| TMA | 160 | $-$\|.25 | 50 EP. | 0 | - | - | - | - | 270 |

Table 1: Hyperparameters of the investigated SPRL algorithms in the different learning environments, as described in appendix D. Note the differentiation between $\epsilon$ for G- and NP-SPRL ($\epsilon_{\mathrm{KL}}$) and WB-SPRL ($\epsilon_{\mathrm{M}}$).

where $T_{\#}\tilde{q}$ is the push-forward of $\tilde{q}$ by $T$, i.e. the application of the weighted Monge maps to the particles of $\tilde{q}$

$$T_{\#}\tilde{q}(\mathbf{c}) = \tfrac{1}{N}\textstyle\sum_{n=1}^{N}\delta_{T(\mathbf{c}_n^q)}(\mathbf{c}).$$

The benefit of this approximation is that the maps $T_{\tilde{q}\to\tilde{p}_J}$ and $T_{\tilde{q}\to\tilde{\mu}}$ only need to be computed once in order to compute $\tilde{p}_{\boldsymbol{\beta}}(\mathbf{c})$ for arbitrary $\boldsymbol{\beta}$. This fast computation of $\tilde{p}_{\boldsymbol{\beta}}(\mathbf{c})$ allows to quickly optimize $\boldsymbol{\beta}$ via objective (7).
To make the optimization of $\tilde{p}_{\boldsymbol{\beta}}(\mathbf{c})$ even faster, we replace the two Wasserstein distances in (7) by approximations, resulting in

$$\min_{w_\mu, w_J} \sum_{n=1}^{N} \|T_{\tilde{q}\to\tilde{\mu}}(\mathbf{c}_n^q) - \mathbf{c}_n^{p_{\boldsymbol{\beta}}}\|^2 \ \ \text{s.t} \ \ \frac{1}{N}\sum_{n=1}^{N} J(\pi, \mathbf{c}_n^{p_{\boldsymbol{\beta}}}) \geq \delta \quad \frac{1}{N}\sum_{n=1}^{N} \|\mathbf{c}_n^{p_{\boldsymbol{\beta}}} - \mathbf{c}_n^q\|^2 \leq \epsilon.$$

The objective term of above constrained optimization problem is a computationally cheap replacement for $\mathcal{W}_2(p_{\boldsymbol{\beta}}, \mu)$. Instead of computing the Wasserstein distance to $\mu$, we simply ask the transported particles of $\tilde{p}_{\boldsymbol{\beta}}$ to follow the transport plan $T_{\tilde{q}\to\tilde{\mu}}$ as closely as possible. This avoids the (comparatively) expensive computation of $\mathcal{W}_2(p_{\boldsymbol{\beta}}, \mu)$ for each choice of $\boldsymbol{\beta}$. Finally, we also avoid to compute $\mathcal{W}_2(p_{\boldsymbol{\beta}}, q)$ but rather replace it with a pessimistic substitute $\frac{1}{N}\sum_{n=1}^{N} \|\mathbf{c}_n^{p_{\boldsymbol{\beta}}} - \mathbf{c}_n^q\|^2 \leq \epsilon$. This substitute is guaranteed to be larger than $\mathcal{W}_2(p_{\boldsymbol{\beta}}, q)^2$ since when using the particle representation, the Wasserstein distance $\mathcal{W}_2(p_{\boldsymbol{\beta}}, q)$ corresponds to finding a permutation $\pi : [1, N] \mapsto [1, N]$ that optimizes

$$\min_\pi \sqrt{\tfrac{1}{N}\textstyle\sum_{n=1}^{N} \|\mathbf{c}_n^{p_{\mathcal{W}}} - \mathbf{c}_{\pi(n)}^q\|^2}.$$

Note that the approximations to the Wasserstein distances in (7) do not alter the transport plan and with that $\tilde{p}_{\boldsymbol{\beta}}(\mathbf{c})$. The approximations may simply alter the optimal parameters $\boldsymbol{\beta}^*$ to (7). However, since the approximations have the same asymptotic behavior, these differences do not hinder good learning performance as can be seen in the experimental section. For the computations of the Monge maps, we use the GeomLoss library (Feydy et al., 2019) which allows to solve entropy regularized optimal transport problems (Cuturi, 2013). We then use the SciPy library (Virtanen et al., 2020) to adjust the weights $\boldsymbol{\beta}$ subject to the performance and distance constraints.

# D    EXPERIMENTAL DETAILS

This section discusses hyperparameters and additional details of the conducted experiments that could not be provided in the main text due to space limitations.

## D.1    ALGORITHM HYPERPARAMETERS

The two main parameters of the three SPRL algorithms are the performance threshold $\delta$ (and $\delta_H$) as well as the allowed distance between subsequent distributions $\epsilon$ (note that we say "distance" as the interpretation is different for WB-SPRL than for G- and NP-SPRL). We did not perform an extensive hyperparameter search for these parameters but proceeded as follows: The performance threshold $\delta$ is chosen such that it is around 50% of the maximally achievable reward and the parameter $\epsilon_M$ is chosen such that around 10 iterations are required to move particles across the whole context space. We then evaluated a larger and a lower value of the parameters and chose the best. For G- and NP-SPRL, we initialized $\epsilon_{\mathrm{KL}}$ with value of 0.05 used in the initial experiments by Klink et al.

However, we realized that larger values slightly improved performance. Additionally, the number of steps per epoch $n_{\text{step}}$ (i.e. between an update of the context distribution) and the number of epochs that take place before any context distribution update is allowed $n_{\text{offset}}$ need to be chosen. In our experiments, the $n_{\text{step}}$ parameter is chosen such that at least 20 episodes are completed per context distribution update. This seemed enough for the two- and one-dimensional context spaces considered. For higher-dimensional context spaces, we assume that it is advisable to either increase the number of completed episodes or to reduce the trust-region parameter, as the regression problem for the expected performance $J(\pi, \mathbf{c})$ will probably require more data. The $n_{\text{offset}}$ parameter mostly serves to give the policy $\pi$ some time to become proficient on easy tasks and the estimator of $J(\pi, c)$ to get somewhat accurate. Nonetheless, SPRL can also works without this offset, as can be seen in the TMA benchmark. Hence, $n_{\text{offset}}$ can be rather seen as a "safety" parameter that interacts with the trust-region parameters $\epsilon_M$ and $\epsilon_{\text{KL}}$ as well as $n_{\text{step}}$.

For G-SPRL, Klink et al. employed a lower bound on the standard deviation $\boldsymbol{\sigma}_{\text{lb}}$ of the context distribution that needs to be respected until the KL divergence w.r.t. $\mu(\mathbf{c})$ falls below a threshold $D_{\text{KL}}$. This avoids a collapse of the context distribution in the case of a very narrow target context distribution $\mu(\mathbf{c})$. In NP-SPRL, we translated this into a lower bound on the entropy of the context distribution. For WB-SPRL, this lower bound is not required. Only, the additional performance lower bound $\delta_H > \delta$ for computing the "value distribution" $p_J(\mathbf{c})$. Aforementioned parameters are listed in Table 1 for the different environments.

For ALP-GMM, the relevant hyperparameters are the percentage of random samples drawn from the context space $p_{\text{rand}}$, the number of completed learning episodes between the update of the context distribution $n_{\text{rollout}}$ as well as the maximum buffer size of past trajectories to keep $s_{\text{buffer}}$. Similar to Klink et al. (2021), we chose them by a grid-search over $(p_{\text{rand}}, n_{\text{rollout}}, s_{\text{buffer}}) \in \{0.1, 0.2, 0.3\} \times \{50, 100, 200\} \times \{500, 1000, 2000\}$ for the maze and pick and place environment. For the point-mass environment, we took the values from Klink et al. (2021).

For GOALGAN, we again took the parameters from Klink et al. (2021) for the point-mass environment and tuned the amount of random noise that is added on top of each sample $\delta_{\text{noise}}$, the number of policy rollouts between the update of the context distribution $n_{\text{rollout}}$ as well as the percentage of samples drawn from the success buffer $p_{\text{success}}$ for the maze and pick and place task via a grid search over $(\delta_{\text{noise}}, n_{\text{rollout}}, p_{\text{success}}) \in \{0.025, 0.05, 0.1\} \times \{50, 100, 200\} \times \{0.1, 0.2, 0.3\}$.

Finally, for ACL, the Exp3.S algorithm that ultimately realizes the curriculum requires two hyperparameters to be chosen: the scale factor for the updates of the arm probabilities $\eta$ and the $\epsilon$ parameter of the $\epsilon$-greedy exploration strategy. We combine ACL with the absolute learning progress (ALP) metric also used in ALP-GMM and conduct a hyperparameter search over $(\eta, \epsilon) \in \{0.05, 0.1, 0.2\} \times \{0.01, 0.025, 0.05\}$. In the discrete pick and place (or discretized maze and point-mass) environment, the absolute learning progress in a context $\mathbf{c}$ can be estimated by keeping track of the last reward obtained in $\mathbf{c}$ and then computing the absolute difference between the return obtained from the current policy execution and the stored last reward. Implementing the ACL algorithm by Graves et al. (2017), we had numerical issues due to the normalization of the ALPs via quantiles computed from a representative sample. This is probably due to the sparse nature of the task. Consequently, we normalized via the maximum and minimum ALP seen over the entire history of tasks.

For PLR, the staleness coefficient $\rho$, the score temperature $\beta$ as well as the replay probability $p$ need to be chosen. We did a grid-search over $(\rho, \beta, p) \in \{0.15, 0.3, 0.45\} \times \{0.15, 0.3, 0.45\} \times \{0.55, 0.7, 0.85\}$ and chose the best configuration for each environment.

For VDS, the parameters for the training of the $Q$-function ensemble, i.e. the learning rate lr, the number of epochs $n_{\text{ep}}$ and the number of minibatches $n_{\text{batch}}$, need to be chosen. Just as for PLR, we conducted a grid-search over $(\text{lr}, n_{\text{ep}}, n_{\text{batch}}) \in \{10^{-4}, 5 \times 10^{-4}, 10^{-3}\} \times \{3, 5, 10\} \times \{20, 40, 80\}$. We chose the same number of environment steps between training of the $Q$-function as for the SPRL algorithms. The parameters of all employed baselines are given in tables 2 and 3.

### D.2 TASK DESCRIPTIONS

We now detail details of the individual experiments, such as context-, state- and action spaces as well as the employed RL algorithms. As RL agents, we use PPO and SAC implemented in the `Stable Baselines` library (Hill et al., 2018). In all tasks, we represent policy- and/or value function networks using neural networks with three hidden layers of 128 neurons and `tanh` activations.

|  | ALP-GMM | | | GOALGAN | | | ACL | |
| --- | --- | --- | --- | --- | --- | --- | --- | --- |
| Env. | $p_{\text{RAND}}$ | $n_{\text{ROLLOUT}}$ | $s_{\text{BUFFER}}$ | $\delta_{\text{NOISE}}$ | $n_{\text{ROLLOUT}}$ | $p_{\text{SUCCESS}}$ | $\eta$ | $\epsilon$ |
| Maze | .2 | 100 | 500 | .1 | 100 | .3 | 0.01 | 0.2 |
| Point-Mass | .2 | 100 | 500 | .1 | 200 | .3 | 0.01 | 0.2 |
| PnP | .2 | 50 | 500 | .1 | 50 | .2 | .05 | .05 |

Table 2: Hyperparameters of the investigated baseline algorithms in the different learning environments, as described in appendix D.

### D.2.1 Maze

The maze task is simulated using MuJoCo (Todorov et al., 2012) by defining a sphere with radius .15 that can move along two prismatic joints along $x$- and $y$-direction. The goal (i.e. the context) can be chosen within $\mathcal{C} = [-3, 15] \times [-3, 15] \subseteq \mathbb{R}$. The actually reachable space of positions (and with that goals) is a subset of $[-1, 13] \times [-1, 13]$ due to the "hole" because of the inner walls of the maze. The state of the environment is given by the $x$- and $y$-position and -velocity of the agent. The reward is sparse, only returning a reward of one if the goal is reached and zero otherwise. A goal is considered reached if the Euclidean distance between goal and position of the point-mass falls below 0.15.

We use the PPO algorithm for learning in this task, updating the policy every 10.000 environment steps for 10 epochs with 20 mini-batches. The parameter $\lambda$ for the generalized advantage estimation is set to 0.995 and the entropy coefficient is set to zero. All other parameters are set to the default of the `Stable Baselines` implementation.

For algorithms that support the specification of an initial context distribution (SPRL variants and GOALGAN), we initialize the context distribution to be uniform over $[-1, 1] \times [-1, 1]$ for GOALGAN, NP-SPRL and WB-SPRL and $\mathcal{N}(\mathbf{0}, .06\mathbf{I})$ for G-SPRL. This initial context distribution samples goals around the initial position of the agent.

### D.3 Point-Mass

The environment setup is the same as the one investigated by Klink et al. (2020b; 2021) with the only difference in the target context distributions $\mu_1(\mathbf{c})$ and $\mu_2(\mathbf{c})$. We use PPO with 4.096 steps per policy update, four update epochs with 8 mini-batches in each policy update, $\lambda = 0.99$ and an entropy coefficient of zero. All other parameters are left to the implementation defaults of the `Stable Baselines` implementation.

### D.4 Pick and Place

The investigated environment is a slightly modified version of the `FetchPickAndPlace` environment from the `OpenAI Gym` Brockman et al. (2016) environment suite. The dynamics and reward functions are exactly the same as the ones from the original environment. We only modify the initial position of the cube as well as the goal to which the cube needs to be moved. The initial $x$- and $y$-position of the cube on the table is sampled uniformly within $[1.31, 1.37] \times [0.72, 0.78]$ and the goal position is sampled within $[1.31, 1.37] \times [0.87, 0.93] \times [0.59, 0.65]$. If the environment is reset to a state in which the cube is in contact with the gripper, its position is not randomized as otherwise the cube may e.g. fall out of the gripper due to randomization, which would counteract the idea behind the curriculum.

|  | PLR | | | VDS | | |
| --- | --- | --- | --- | --- | --- | --- |
| Env. | $\rho$ | $\beta$ | $p$ | LR | $n_{\text{EP}}$ | $n_{\text{BATCH}}$ |
| Maze | .3 | .3 | .7 | $5 \times 10^{-4}$ | 10 | 40 |
| Point-Mass | .3 | .3 | .7 | $5 \times 10^{-4}$ | 3 | 40 |
| PnP | .3 | .45 | .7 | $5 \times 10^{-4}$ | 10 | 40 |

Table 3: Hyperparameters of the investigated baseline algorithms in the different learning environments, as described in appendix D.

The demonstration is generated from a simple controller that first aligns the gripper with a position 10cm above the cube, then lowers the gripper until the cube is between its two fingers, closes the gripper and then moves the endeffector to the goal position. The control command for the robot endeffector is computed via

$$\mathbf{u} = \text{sign}(\Delta_{\mathbf{x}}) \max(0.2, |\Delta_{\mathbf{x}}|), \quad \Delta_{\mathbf{x}} = \mathbf{x}_{\text{des}} - \mathbf{x},$$

where $\mathbf{x}_{\text{des}}$ is the desired- and $\mathbf{x}$ the current endeffector position. Lower bounding the control command by $0.2$ avoids the movements becoming very small as the endeffector approaches the target position, which would lead to unnecessarily long demonstrations. As soon as we are within a $0.01$ distance to the target position we switch to the next step in the previously outlined sequence.

As an RL agent, we use SAC with $5$ steps per policy update, a batch size of $512$, a maximum replay buffer size of $100.000$ and a learning rate $3 \times 10^{-4}$. All other parameters are left to the implementation defaults of the `Stable Baselines` implementation.

As in the maze task, we specify initial context distributions for GOALGAN and the investigated SPRL variants. These context distributions assign uniform probability to the last $4$ time-steps of the trajectory.

