# OpenReview forum: "Metrics Matter: A Closer Look on Self-Paced Reinforcement Learning"
_ICLR.cc/2022/Conference — ICLR 2022 Submitted_

### Official Review · Reviewer_w8rz · 2021-10-29

**Correctness:** 3
**Technical Novelty And Significance:** 2
**Empirical Novelty And Significance:** 2
**Recommendation:** 6
**Confidence:** 4

**Main Review:**

### Correctness

The paper clearly states the algorithm, intended changes and their purpose as well the experimental details. The change in metric itself makes sense, as demonstrated in the paper, to increase the range of problems to which self-paced reinforcement learning (SPRL) can be applied. The experiments not only show the increased performance using the curriculum update, but also explore how the curriculum itself develops.

### Technical Novelty And Significance

The paper is incremental by extending SPRL by changing the metric. So, it’s a combination of two existing ideas (which totally makes sense), but the overall concept of the SPRL algorithm stays the same. Although the experiments show a clear performance improvement, the relevance of the results are not fully clear and therefore also not the significance of the proposed SPRL modification; see below for more details.

### Empirical Novelty And Significance

The authors show extended use cases for the SPRL algorithm and use appropriate baselines, although the experimental scope is still quite limited compared to other curriculum learning papers in RL. Even though target distributions can take arbitrary shapes now, the authors stick to very narrow targets and do not discuss generalization beyond simple bimodal distributions. This limits comparability with other methods and ultimately the significance of the paper’s contributions.

### General comments

Sections 5.1 and 5.3 in particular highlight the need for non-parametric target distributions. The PointMass experiment in 5.2, however, was not as convincing as it does not show how well the Wasserstein barycenters (WB-SPRL) approach scales in general. If the target distribution can now be bimodal instead of unimodal, it is still an improvement, but perhaps not as interesting as if 10 or more different gate configurations would be possible. I find this particularly relevant as this is the only experiment in which the naive discretization (NP-SPRL) looks to perform significantly worse than WB-SPRL. Would it be fair to say that for unimodal cases, there is no large performance gap between the two?
I would rate the overall significance of these findings much higher if WB-SPRL can be expected to perform well beyond the bimodal case. Do you have any intuition on the limitations for context distributions for WB-SPRL?


On a similar line of thoughts, I wonder about the experimental setting. For example, in the experiments on Maze (Section 5.1), the context c was able to encode an infeasible environment. Thinking about real-world applications, I would expect that only feasible contexts and environments are available for training, especially given that we assume a fairly good understanding of the context space (we already know quite a bit about instance difficulties, for example). Furthermore, I believe that the target distribution \mu should be fairly broad in many cases, since many possible environments have to be covered by a generalizing agent. The target distribution in the paper at hand are all fairly narrow, which might benefit SPRL’s performance compared to other baselines, but also limit the algorithm in practice. Could the authors please comment on the real-world applicability of their SPRL variant in relation to wider distributions of \mu.

I’d also be interested to know more about SPRL’s hyperparameters. While they are stated in the appendix, there is no clear indication to what degree they affect the performance and if this differs between the variations. How were they chosen in the first place? Was there any hyperparameter optimization involved?

The related work on curriculum learning in RL could be improved. There have been many publications with good results in the past years beyond SPRL, two additional papers on the topic from 2018 and 2019 do not reflect the state of the art. I recommend that this section should be extended, even though not all work on CL in RL may have exactly the same focus of solving specific hard instances or generating curricula through generating instances as SPRL. Examples include:
* Automatic Curriculum Learning through Value Disagreement, Zhang et al. https://proceedings.neurips.cc/paper/2020/file/566f0ea4f6c2e947f36795c8f58ba901-Paper.pdf
* Safe Reinforcement Learning via Curriculum Induction, Turchetta et al.
https://proceedings.neurips.cc/paper/2020/file/8df6a65941e4c9da40a4fb899de65c55-Paper.pdf
* Prioritized Level Replay, Jiang et al.
http://proceedings.mlr.press/v139/jiang21b/jiang21b.pdf
* Self-Paced Context Evaluations for Contextual Reinforcement Learning, Eimer et al.
http://proceedings.mlr.press/v139/eimer21a/eimer21a.pdf
* TeachMyAgent: a Benchmark for Automatic Curriculum Learning in Deep RL, Romac et al.
http://proceedings.mlr.press/v139/romac21a.html

Could the authors please also briefly explain in the rebuttal, why there is no empirical comparison against these baselines? Or why a comparison against other baselines made more sense?

### Minor questions and comments:
* Equation 4: Why can we get rid of \delta from Equation 3?
* End of Section 3: Why do you assume 2-Wasserstein distances under an euclidean metric?
* There’s no reference for contextual RL in the related work section. The oldest reference I’m aware would be Hallak et al. 2015
* Did you adapt the baseline hyperparameters at all from their original settings? I do not think the appendix states this explicitly. Did you re-implement them or could you use the original code for all of them?


**Summary Of The Paper:**

The authors extend the Self-Paced Reinforcement Learning, which samples environment instances from a distribution that shifts from an (easy) starting to a (hard) target distribution. The algorithm has previously been shown to enable agents to solve hard environments. Previously, the target distribution was limited to a Gaussian, which the authors now extend using Wasserstein barycenters. This extension allows more elaborate target distributions, e.g. expert demonstrations.

**Summary Of The Review:**

The paper proposes a well-motivated extension to the SPRL algorithm and demonstrates its usefulness in interesting experiments. The significance of these results is somewhat unclear, though, as the naive context discretization seems to perform just as well as the proposed metric in every but a very limited case. Additionally, the nature of the work is rather incremental with limited impact and novelty.

---

> ### Author Response · Authors · 2021-11-20
> **Answer to Reviewer w8rz**
>
> We thank the reviewer for the baseline suggestions and thoughtful comments. We discuss them below:
>
> > Even though target distributions can take arbitrary shapes now, the authors stick to very narrow targets and do not discuss generalization beyond simple bimodal distributions. This limits comparability with other methods and ultimately the significance of the paper’s contributions.
>
> We added an evaluation of WB-SPRL in the TeachMyAgent benchmark [TMA] to compare it in a different scenario than the maze task, the point-mass and the pick and place task. We hope that this additional evaluation helps readers to situate the method better in the field of curriculum RL. We additionally added more recent baselines [VDS] and [PLR] (see our answer at the bottom).
>
> Regarding the (narrow) target distributions, there seems to be a misunderstanding. The target distribution of the maze task is a uniform distribution. In the updated version of the paper, the [TMA] benchmark is another task with a uniform target distribution.
> Regarding generalization beyond the investigated target distributions, please see our next answer.
>
> [TMA] Romac, Clément, et al. TeachMyAgent: a Benchmark for Automatic Curriculum Learning in Deep, ICML 2021
>
> [VDS] Zhang, Yunzhi, Pieter Abbeel, and Lerrel Pinto. "Automatic curriculum learning through value disagreement." NeurIPS, 2020.
>
> [PLR] Jiang, Minqi, Edward Grefenstette, and Tim Rocktäschel. "Prioritized level replay." ICML, 2021
>
> > If the target distribution can now be bimodal instead of unimodal, it is still an improvement, but perhaps not as interesting as if 10 or more different gate configurations would be possible
>
> We realized that we should have stressed more strongly that the particle-based representation of WB-SPRL is capable of approximating arbitrary target distributions, as long as we can sample those target distributions (which is always assumed in CRL). We added an additional section in the appendix demonstrating the capacity of the particle-based representation. The particular choice of training distributions is rather a result of our aim to use as intuitive experiments as possible that already challenge G- and NP-SPRL.
>
> > I find this particularly relevant as this is the only experiment in which the naive discretization (NP-SPRL) looks to perform significantly worse than WB-SPRL.
>
> There seems to be a misunderstanding, as NP-SPRL also performs poorly in the Pick and Place task. We realized that this has been hard to spot in Figure 8b and changed the plot accordingly. To clarify, NP-SPRL is highly reliant on a well-defined target task distribution. If the log-likelihood function is non-Gaussian or non-Uniform, the interpolation can fail catastrophically. WB-SPRL does not suffer from the subtleties of the task specification.
>
> > I would expect that only feasible contexts and environments are available for training, especially given that we assume a fairly good understanding of the context space (we already know quite a bit about instance difficulties, for example). Furthermore, I believe that the target distribution \mu should be fairly broad in many cases, since many possible environments have to be covered by a generalizing agent.
>
> The presence of infeasible contexts is fairly common in CRL, as e.g. ALP-GMM and GoalGAN are built around these ideas as well. Indeed, the Maze environment with its uniform target distribution over tasks is taken from the GoalGAN experimental section. Also the field of Unsupervised Environment Discovery or Domain Randomization needs to solve a trade-off between high diversification of training tasks and feasibility. The [TMA] benchmark also targets this trade-off.
> Regarding wide and narrow target distributions: We believe that it is interesting to investigate both scenarios, as e.g. [SPRL] demonstrates that a narrow target distribution can arise when generating curricula by lowering precision requirements in robotic scenarios (Ball-in-a-Cup task in Section 6.2.3). As already mentioned, we performed an additional experiment with WB-SPRL in a wide distribution scenario (the [TMA] benchmark), showing that the use of Wasserstein barycenters allows for an improved performance compared to G-SPRL. Further, the experiments indicate future ways of improving SPRL, such as an adaptive sampling of the context distribution in the style of ALP-GMM.
>
> [TMA] Romac, Clément, et al. TeachMyAgent: a Benchmark for Automatic Curriculum Learning in Deep, ICML 2021
>
> [SPRL] Klink, Pascal, et al. "A Probabilistic Interpretation of Self-Paced Learning with Applications to Reinforcement Learning. JMLR, 2021

---

> > ### Author Response · Authors · 2021-11-20
> > **Answer to Reviewer w8rz**
> >
> > > I’d also be interested to know more about SPRL’s hyperparameters. While they are stated in the appendix, there is no clear indication to what degree they affect the performance and if this differs between the variations. How were they chosen in the first place? Was there any hyperparameter optimization involved?
> >
> > We added a description on how the hyperparameters for SPRL were chosen in the experiments. The most important hyperparameters are the desired performance threshold as well as the trust-region for the context distribution update. Especially for WB-SPRL both parameters allow for an intuitive way of choosing them.
> >
> > As visible in the appendix, the performance thresholds are chosen to be around half of the maximally achievable reward (the maximum performance in the point-mass environment is around 8-9 and the maximum success rate for Maze and PickAndPlace is 1). We then performed experiments with a higher and lower value, then choosing the better.
> >
> > For WB-SPRL, the trust-region parameter represents the average allowed displacement of each particle per context distribution update. We chose the value of this trust-region such that around 10 iterations are needed to move a particle fully through the task space, again evaluating variations with a higher and lower value and choosing the better. For NP-SPRL, the trust-region has a less intuitive explanation and we realized that increasing the value of 0.05 used in the initial experiments by Klink et al. slightly improved performance. Note that for the narrow target tasks (particularly with non-Gaussian likelihoods), the problems of NP-SPRL persisted regardless of the value of the trust-region, as it only changes how fast the interpolation between current and target task distribution progresses, but does not alter the shape of the interpolation.
> >
> > For the baselines, we conducted grid-search to find their parameter values as we detail now in appendix D.1.
> >
> > > The related work on curriculum learning in RL could be improved. There have been many publications with good results in the past years beyond SPRL, two additional papers on the topic from 2018 and 2019 do not reflect the state of the art. I recommend that this section should be extended, even though not all work on CL in RL may have exactly the same focus of solving specific hard instances or generating curricula through generating instances as SPRL.
> >
> > We agree that the related work should have been more extensively discussed to avoid some of the questions of the reviewers. The revised version now better situates our method in the literature and also better narrows down the scope of our investigation.
> >
> > > Could the authors please also briefly explain in the rebuttal, why there is no empirical comparison against these baselines? Or why a comparison against other baselines made more sense?
> >
> > This seems to be a misunderstanding, as we compare to methods benchmarked in [TMA]: ALP-GMM and GoalGAN. ALP-GMM is a particularly strong baseline judging from the results in [TMA] and our own experimental results. Furthermore, it is a representative for many related methods such as [SPACE], [PLR], or [VDS], which all introduce some proxy (Learning Progress, Regret, Q-Ensemble disagreement) whose expectation over the chosen task distribution should be maximized. To address the concerns regarding adequateness of the chosen baselines, we further added comparisons to [PLR] and [VDS] in our paper. We do not consider [SRL] in our work, as the method assumes a small set of training tasks (in their paper called interventions) to choose from in order to generate a curriculum via Bayesian Optimization. Adapting such an approach to continuous goal spaces seemed a work on its own.
> >
> > [TMA] TeachMyAgent: a Benchmark for Automatic Curriculum Learning in Deep RL, Romac et al. http://proceedings.mlr.press/v139/romac21a.html
> >
> > [SPACE] Self-Paced Context Evaluations for Contextual Reinforcement Learning, Eimer et al. http://proceedings.mlr.press/v139/eimer21a/eimer21a.pdf
> >
> > [PLR] Prioritized Level Replay, Jiang et al. http://proceedings.mlr.press/v139/jiang21b/jiang21b.pdf
> >
> > [VDS] Automatic Curriculum Learning through Value Disagreement, Zhang et al. https://proceedings.neurips.cc/paper/2020/file/566f0ea4f6c2e947f36795c8f58ba901-Paper.pdf
> >
> > [SRL] Safe Reinforcement Learning via Curriculum Induction, Turchetta et al. https://proceedings.neurips.cc/paper/2020/file/8df6a65941e4c9da40a4fb899de65c55-Paper.pdf

---

> > > ### Comment · Reviewer_w8rz · 2021-11-23
> > > **Thanks for the thorough reply**
> > >
> > > Hi,
> > >
> > > Thank you very much for the thorough reply.
> > > I have two further comments regarding the reply.
> > >
> > > > For the baselines, we conducted grid-search to find their parameter values as we detail now in appendix D.1.
> > >
> > > It's good that you also optimized the baselines, but grid search is definitely not a very efficient way of doing it. I would recommend to use at the very least random search, or even better Bayesian Optimization.
> > > Furthermore, it would be good if you could ensure that you invested the same amount of tuning of the baselines as used for your approach.
> > >
> > > > ALP-GMM is a particularly strong baseline judging from the results in [TMA] and our own experimental results. Furthermore, it is a representative for many related methods such as [SPACE], [PLR], or [VDS], which all introduce some proxy (Learning Progress, Regret, Q-Ensemble disagreement) whose expectation over the chosen task distribution should be maximized.
> > >
> > > I'm not sure whether I would agree to the statement of being representative. In particular, not all of them make use of a teacher which is a quite important distinguishing factor.

---

### Official Review · Reviewer_5LxC · 2021-11-02

**Correctness:** 3
**Technical Novelty And Significance:** 3
**Empirical Novelty And Significance:** 2
**Recommendation:** 5
**Confidence:** 3

**Main Review:**

### Strengths

1) This paper tackles an important problem, namely that of developing principled automatic curricula for RL.

2) The experiments are well designed and provide useful insights into the learning dynamics and properties of different methods. I particularly liked the analysis showing how the context distributions change throughout training, as well as the interpolation between different distributions, for G-SPRL and NP-SPRL.

3) The paper is clear overall and I found the claims regarding the limitations of NP-SPRL and G-SPRL relative to SPRL to be well supported by the experiments. However, I believe a few of the empirical claims regarding WB-SPRL are not as well motivated (see more details below).

4) Even if the proposed method is a natural extension of SPRL using the Wasserstein metric, this hasn't been done before (as far as I know) and it is well motivated, making it a reasonable contribution at ICLR (in terms of novelty).

### Weaknesses

1) While this paper makes a reasonably convincing case for using WB-SPRL rather than G-SPRL or NP-SPRL, especially when the distribution of contexts isn't Gaussian and the space is high-dimensional, it doesn't convincingly show or explain the benefits of using WB-SPRL over other curriculum-based methods. In particular, ALP-GMM is just as good on two out of the three tasks used for evaluation and almost as good on the remaining one. In addition, the paper doesn't contain an in depth discussion of how WB-SPRL compares with other curriculum-based methods and when one can expect it to work better or why one may prefer to use this method (rather than e.g. ALP-GMM) even if it performs just as well.

2) Most of the evaluation domains used (except for PNP) seem somewhat contrived as if they were designed for NP-SPRL and G-SPRL to fail and WB-SPRL to perform well. Thus, it would be useful to see how these methods perform on more realistic / natural domains e.g. other robotics tasks similar to PNP.

3) Evaluating the methods on other tasks (e.g. other configurations of Maze or even a setting where both the maze layout and the goal location are changing, different configurations of Point Mass as used in Florensa et al. 2019, or other more realistic robotic tasks), and comparing WB-SPRL with other effective curriculum-based methods (e.g. Asymmetric Self-Play by Sukhbaatar et al. 2017 which outperforms GoalGAN on certain tasks or an adaptation of AMIGo by Campero et al. 2020 to continuous control domains), would help better motivate the use of the proposed approach and shed light on when this method can help the most.

4) The paper seems to be missing experiments with ACL-GMM on Maze and Point Mass. Given that ACL-GMM is the best method on PNP, it would be useful to see how it performs on the other domains as well to better understand whether WB-SPRL provides additional gains on those settings.

5) This paper also misses mentions to a lot of the relevant literature on curriculum learning, intrinsic motivation, exploration, or other approaches for solving challenging / sparse reward RL tasks. I found the related work section to be too narrow, brief, and lacking the necessary context to understand the paper's contribution. A discussion of how SPRL and in particular WB-SPRL compares with other approaches is also needed.

### Questions / Requests
1) Could you provide experiments showing how the performance of NP-SPRL scales with the resolution of the discretization? Additional analysis showing the $p_{\alpha, \eta}(c)$ at different iterations would also be valuable.
2) Do you have any intuition of why ALP-GMM performs just as well as WB-SPRL on Maze? And why does ALP-GMM perform worse on Point Mass?
3) In Figures 4 and 5, it would be useful to add the insights drawn from these experiments in the caption (for completeness and ease of read).
4) In Figure 8a, can you show p(t) for the other approaches (e.g. G-SPRL, NP-SPRL)? Can you explain what the colors mean and also add this information in the caption?
5) Do you have any intuition of why GoalGAN doesn't work as well, especially on Maze and Point Mass, since these domains are very similar to the ones used in the original paper?
6) When can we expect WB-SPRL to provide a benefit over other curriculum-based approaches such as ALP-GMM or GoalGAN? Or why should one choose to use this method over the alternatives?
7) Can you include the goal density / context distributions of GoalGAN,ALP-GMM, and WB-SPRL for comparison i.e. include them Figure 2 in order to better understand how the learning dynamics / learned behaviors differ from other SPRL variants?


**Summary Of The Paper:**

This paper investigates self-paced reinforcement learning (SPRL) which is a type of curriculum-based RL. The authors empirically demonstrate the (different) limitations of a few SPRL variants (one which uses a Gaussian approximation, G-SPRL, and a non-parametric one, NP-SPRL). They propose the use of a Wasserstein metric instead of the KL divergence (WB-SPRL) and show that this leads to higher performance, more desirable behaviors, and more meaningful interpolation between different MDPs / contexts, on three continuous control tasks. The authors also compare their approach with other automatic curriculum algorithms.

**Summary Of The Review:**

Overall, I think this paper provides a good understanding of the different SPRL variants, their limitations, and proposes a new method which provides some empirical benefits (over other SPRL methods) and is theoretically motivated.

My main concerns are the incomplete related work section and the limited benefits demonstrated by WB-SPRL over other curriculum-based methods. Hence, I believe the paper needs some more work to warrant acceptance at ICLR.

I'm willing to increase the score if the authors include references to the relevant works from the literature, a discussion of how WB-SPRL compares with other curriculum RL methods (theoretically and empirically), and include more experiments that (more convincingly) demonstrate WB-SPRL's benefits over other approaches (with some intuition of why / in which settings we can expect WB-SPRL to be better).

---

> ### Author Response · Authors · 2021-11-20
> **Answer to Reviewer 5LxC**
>
> We thank the reviewer for the constructive feedback and now outline how we incorporated it:
>
> > While this paper makes a reasonably convincing case for using WB-SPRL rather than G-SPRL or NP-SPRL, especially when the distribution of contexts isn't Gaussian and the space is high-dimensional, it doesn't convincingly show or explain the benefits of using WB-SPRL over other curriculum-based methods. In particular, ALP-GMM is just as good on two out of the three tasks used for evaluation and almost as good on the remaining one. In addition, the paper doesn't contain an in depth discussion of how WB-SPRL compares with other curriculum-based methods and when one can expect it to work better or why one may prefer to use this method (rather than e.g. ALP-GMM) even if it performs just as well.
>
> Reading the reviewer comments, we realized that the motivation behind the paper may have been suboptimally described in the current version of the paper and we adjusted it by extending the related work section to better contrast it to other approaches.
> In our opinion, SPRL is an interesting approach to curriculum RL as it aims to formulate the curriculum generation as an interpolation between task distributions, which e.g. explicitly connects it to the concept of Homotopic Continuation Methods [HOM] in the field of optimization/nonlinear systems of equations. While this connection is commonly used as a motivation for curricula in machine learning, empirically successful methods like ALP-GMM or GoalGAN are unfortunately missing this explicit link in their formulation. Nonetheless, these methods are definitely successful and the goal of this paper is not to beat these methods, but to improve SPRL while retaining the explicit link to continuation methods.
> From a practical perspective, the main benefit of SPRL is the typically strong performance when aiming to solve a particular hard target task. This has been shown in the work by Klink et al. [SPRL] and also in our Point-Mass environment, where ALP-GMM learns less reliable policies compared to WB-SPRL. To better contrast our method to existing ones - particularly the successful ALP-GMM - we evaluated WB-SPRL in the “TeachMyAgent” benchmark [TMA] (proposed by the authors of ALP-GMM). We see that WB-SPRL improves upon G-SPRL (which has already been evaluated in this benchmark in [TMA]). Further, the results indicate that SPRL may still be improved by incorporating prioritized sampling of the context distribution, as e.g. performed by ALP-GMM.
>
> [HOM] Allgower, Eugene L., and Kurt Georg. Numerical continuation methods: an introduction. Vol. 13. Springer Science & Business Media, 2012.
>
> [TMA] Romac, Clément, et al. TeachMyAgent: a Benchmark for Automatic Curriculum Learning in Deep, ICML 2021
>
> [SPRL] Klink, Pascal, et al. "A Probabilistic Interpretation of Self-Paced Learning with Applications to Reinforcement Learning. JMLR, 2021
>
> >Most of the evaluation domains used (except for PNP) seem somewhat contrived as if they were designed for NP-SPRL and G-SPRL to fail and WB-SPRL to perform well. Thus, it would be useful to see how these methods perform on more realistic / natural domains e.g. other robotics tasks similar to PNP.
>
> We agree that the experiments challenge NP- and G-SPRL, as our goal with this paper was to highlight the shortcoming of the KL-based interpolation between task distributions and propose an alternative that performs more robustly. However, none of the investigated environments has been customly created for that purpose. The point-mass environment has been already investigated in the seminal works on [SPRL] and the maze task has been investigated by [GoalGAN]. As already mentioned, we additionally performed experiments in the [TMA] benchmark, where we show clear improvements over G-SPRL, sometimes reaching top performance across methods. We hope that this additional investigation both makes the experiments seem less contrived and helps to better situate SPRL in the domain of curriculum RL.
>
> [SPRL] Klink, Pascal, et al. "A Probabilistic Interpretation of Self-Paced Learning with Applications to Reinforcement Learning. JMLR, 2021
>
> [GoalGAN] Florensa, Carlos, et al. "Automatic goal generation for reinforcement learning agents." ICML, 2018.

---

> > ### Author Response · Authors · 2021-11-20
> > **Answer to Reviewer 5LxC**
> >
> > > Evaluating the methods on other tasks (e.g. other configurations of Maze or even a setting where both the maze layout and the goal location are changing, different configurations of Point Mass as used in Florensa et al. 2019, or other more realistic robotic tasks), and comparing WB-SPRL with other effective curriculum-based methods (e.g. Asymmetric Self-Play by Sukhbaatar et al. 2017 which outperforms GoalGAN on certain tasks or an adaptation of AMIGo by Campero et al. 2020 to continuous control domains), would help better motivate the use of the proposed approach and shed light on when this method can help the most.
> >
> > We performed additional experiments in the stump-track task of the [TMA] benchmark and further included additional baselines (proposed by reviewer w8rz) in our evaluations: [PLR] and [VDS]. The additional baselines underline the benefit of SPRL in tasks with known narrow target distributions, as none of the additional baselines matches the performance of WB-SPRL in the point mass task. They further underline our impression that ALP-GMM is a very strong baseline.
> >
> > [TMA] Romac, Clément, et al. TeachMyAgent: a Benchmark for Automatic Curriculum Learning in Deep, ICML 2021
> >
> > [VDS] Zhang, Yunzhi, Pieter Abbeel, and Lerrel Pinto. "Automatic curriculum learning through value disagreement." NeurIPS, 2020.
> >
> > [PLR] Jiang, Minqi, Edward Grefenstette, and Tim Rocktäschel. "Prioritized level replay." ICML, 2021
> >
> > >The paper seems to be missing experiments with ACL-GMM on Maze and Point Mass. Given that ACL-GMM is the best method on PNP, it would be useful to see how it performs on the other domains as well to better understand whether WB-SPRL provides additional gains on those settings.
> >
> > We are sorry for this inconsistency and resolved it by adding the additional comparisons to the paper. The reason why we only evaluated ACL in the PNP task is because the ACL algorithm is formulated as a bandit problem. The PNP task is well-suited for this bandit formulation due to its rather small number of discrete contexts. In the continuous settings, the discretized state space has a much higher number of discrete contexts and hence bandit arms. Consequently, we did not expect ACL to perform as well in the continuous settings. This assumption is confirmed by the experimental results, where ACL never performs best and performs poorly in the point mass environment, in which training for particular target tasks is required.
> > This paper also misses mentions to a lot of the relevant literature on curriculum learning, intrinsic motivation, exploration, or other approaches for solving challenging / sparse reward RL tasks. I found the related work section to be too narrow, brief, and lacking the necessary context to understand the paper's contribution. A discussion of how SPRL and in particular WB-SPRL compares with other approaches is also needed.
> > We adjusted the related work section to better map out related works and -fields and to better explain our particular contribution in the field of curriculum RL. We believe that the scope of our work is now much better represented and are very happy for this feedback.
> >
> > > Could you provide experiments showing how the performance of NP-SPRL scales with the resolution of the discretization? Additional analysis showing the $p_{\alpha, \eta}(\mathbf{c})$ at different iterations would also be valuable.
> >
> > Due to our focus on providing more baseline results during the revision phase, we unfortunately did not have time to conduct such an investigation. However, it is definitely an interesting further evaluation.
> >
> > > Do you have any intuition of why ALP-GMM performs just as well as WB-SPRL on Maze? And why does ALP-GMM perform worse on Point Mass?
> >
> > The benefit of WB-SPRL on the point mass task is the notion of target distribution that ALP-GMM is missing compared to WB-SPRL. ALP-GMM (and other CRL methods that generate a prioritized sampling), typically assume a uniform distribution over the context space. This reduces their performance if this assumption is not valid (as for the point mass environment).
> >
> > > In Figure 8a, can you show $p(t)$ for the other approaches (e.g. G-SPRL, NP-SPRL)? Can you explain what the colors mean and also add this information in the caption?
> >
> > Thank you for pointing out this missing information in the caption of Figure 8: We revised the caption of the figure to explain the color coding. The curricula generated by NP-SPRL are shown in Appendix B.

---

> > > ### Author Response · Authors · 2021-11-20
> > > **Answer to Reviewer 5LxC**
> > >
> > > > Do you have any intuition of why GoalGAN doesn't work as well, especially on Maze and Point Mass, since these domains are very similar to the ones used in the original paper?
> > >
> > > For the point-mass environment, again the lack of notion of target distribution is giving WB-SPRL an advantage that probably causes the poor performance. For the Maze environment, we were ourselves surprised regarding the performance. We reviewed the implementation from Klink et al and performed a grid search over hyperparameters (as mentioned in the appendix). But the gap in performance w.r.t. ALP-GMM, WB-SPRL and PLR persisted.
> > >
> > > > When can we expect WB-SPRL to provide a benefit over other curriculum-based approaches such as ALP-GMM or GoalGAN? Or why should one choose to use this method over the alternatives?
> > >
> > > As we aimed to motivate throughout the paper. SPRL is an interesting approach as it aims to formulate the curriculum as an interpolation between task distributions, which is different from the approach of many other methods that focus on maximizing a metric (such as learning progress) to find the curriculum. The SPRL formulation is particularly beneficial when performing interpolation to a distribution of target tasks that is significantly "smaller" than the context space. However, as we show in our experiments, it can also reach similar performance to top-performing methods like ALP-GMM in other scenarios. As we argue at the end of Section 6, it may indeed be possible to combine the SPRL framework with such methods to combine the benefits of both.
> > >
> > > > Can you include the goal density / context distributions of GoalGAN,ALP-GMM, and WB-SPRL for comparison i.e. include them Figure 2 in order to better understand how the learning dynamics / learned behaviors differ from other SPRL variants?
> > >
> > > As previously mentioned, we for now focused on the more severe concerns of the reviewers particularly w.r.t. baselines. But we will definitely take this suggestion into account for a future iteration of the paper.

---

> > > > ### Comment · Reviewer_5LxC · 2021-11-21
> > > > **Thank you for the response**
> > > >
> > > > Thank you for the response. While I appreciate the authors' efforts in improving the related work section, adding more baselines, and evaluating the method on an additional task, I believe the main points have not been adequately addressed.
> > > >
> > > > My main concern is that the empirical results do not make a strong case for using the proposed approach, relative to other curriculum based methods. While I understand the theoretical justification, if these properties are indeed relevant, you should be able to empirically show the benefits of this method, at least in some special cases (and clearly explain which those are). Otherwise, the scope and significance of this work is rather limited.
> > > >
> > > > In addition, the current results only indicate a clear advantage in one out of four tasks i.e. Point-Mass. In the new TMA task, it looks like the gap between WB-SPRL and ALP-GMM is very small and potentially not statistically significant, even on the "Mostly unfeasible task space" (which the authors claim is the one we should care most about and where WB-SPRL should provide the most benefit), while ALP-GMM wins on all the other axes. My conclusion from these experiments is that ALP-GMM is more robust and works well on more settings than WB-SPRL, and even on the setting in which WB-SPRL is expected to provide most gains, the gains are marginal. The results seem to be missing error bars, which makes it difficult to assess their significance.
> > > >
> > > > I was somewhat disappointed to see that the authors didn't take some of the feedback into account. In particular, both myself and reviewer w8rz suggested running experiments on more challenging settings that e.g. go beyond the bimodal distribution and / or use multiple configurations of the goal and maze in the Point-Mass task.
> > > >
> > > > Given the above, I am still leaning towards rejecting the paper.

---

### Official Review · Reviewer_zmCP · 2021-11-02

**Correctness:** 4
**Technical Novelty And Significance:** 3
**Empirical Novelty And Significance:** 3
**Recommendation:** 6
**Confidence:** 2

**Main Review:**

Overall I found this paper to be well written and I understand the issues with prior work it aims to resolve. This being said, I will caveat my review by saying that I am not familiar enough with optimal transport math to be confident in my ability to review the equations and more technical details.

I enjoyed Figure 6 as it illustrated why working with the wasserstein metric would give a smoother interpolation from unimodal to multi-modal distributions.

The goal spaces in this paper all seem rather low dimensional and simple. It would be more promising to me if there were experiments with e.g. harder mazes (more invalid regions). What happens when the starting distribution is "deceiving" — i.e. assume the particle must first move away from the goal down a corridor, will this notion of KL/wasserstein fail because the only reachable states will be in the opposite direction of the target distribution?

Figure 3: Why do all methods underperform the default at the beginning of training? Is it due to the choice of starting distribution?

It looks to me like the margin between paragraphs has been editted. Please revert it to what is expected with the ICLR style.

In the maze task, why does NP-SPRL have non-zero (or closer to zero) mass attributed to the unreachable center part of the maze?

Pick and place is solvable without a recorded demonstration with e.g. hindsight experience replay and no curriculum. It's a bit disappointing that the experiments with this environment are with a recorded trajectory.

**Summary Of The Paper:**

The authors try to alleviate the need for parametric distributions (namely a Gaussian) from the self paced RL framework (SPRL). They seek to shift current distribution over tasks to the target distribution of tasks under 2 constraints: keeping the current distribution close to the target and also that the expected return of the current distribution remain above some threshold. They compare SPRL using a KL divergence metric between a discretized proposal distribution and the target as well as a Wasserstein distance metric between particle based approximations. They show that in a set of toy tasks with low dimensional goal spaces can outperform existing baselines, e.g. GoalGan, as well as the previous instantiation of SPRL with parametric distributions.

**Summary Of The Review:**

I enjoyed the paper; however, because I'm not familiar with the math I will defer to other reviewers on their assessment on this front. I'm giving this paper a 6 because I felt that in the end it did not prove it's new formulation was better but rather showed it experimentally. For a primarily experimental paper, I felt the experimental domains were rather simplistic.

---

> ### Author Response · Authors · 2021-11-20
> **Answer to Reviewer zmCP (1/2)**
>
> We thank the reviewer for the thoughtful comments, which in combination with the other reviews helped us the create an improved version of the paper. In the following, we address the individual questions/concerns.
>
> > The goal spaces in this paper all seem rather low dimensional and simple. It would be more promising to me if there were experiments with e.g. harder mazes (more invalid regions). What happens when the starting distribution is "deceiving" — i.e. assume the particle must first move away from the goal down a corridor, will this notion of KL/wasserstein fail because the only reachable states will be in the opposite direction of the target distribution?
>
> We agree that the task/goal spaces are rather low-dimensional in our setting, as we aim to convey the main concept that generating curricula in the self-paced framework should incorporate a notion of metrics on the task/goal space. Evaluations in higher-dimensional context spaces are a promising part of future work, as the particle-based WB-SPRL seems well-suited for such scenarios. However, even recent benchmarks for automated curriculum generation in continuous spaces, such as [TMA], assume low-dimensional task/goal spaces. We indeed evaluated WB-SPRL in parts of the [TMA] benchmark, showing promising performance and clear improvement over SPDL, and better visualizing the inner workings of our method.
> Regarding deceiving starting distributions: The behaviour of SPRL - and hence whether initialization can lead to failures - depends on the specification of the target task distribution. For a uniform target task distribution, such local optima are not present. For target distributions focusing on narrow parts of the context space, such problems may indeed occur. However, we want to highlight that popular curriculum RL methods (such as all the investigated baselines) implicitly assume a uniform target density and that consequently reasoning over such local optima is only possible in the self-paced framework. This is one of the reasons why we believe that work on SPRL is interesting.
>
> [TMA] Romac, Clément, et al. TeachMyAgent: a Benchmark for Automatic Curriculum Learning in Deep, ICML 2021
>
> > Figure 3: Why do all methods underperform the default at the beginning of training? Is it due to the choice of starting distribution?
>
> The initial underperformance of the curriculum-based training is explained by the narrow target task distribution in this task. Since the curricula first focus on easier tasks, the policies are initially less proficient on the target tasks than a policy directly training on those target tasks. However, we see that only training on the target tasks converges to a poor solution in which the agent simply stays close to the wall (visualization in the appendix).
>
> > It looks to me like the margin between paragraphs has been edited. Please revert it to what is expected with the ICLR style.
>
> We carefully checked our latex document for any changes to the paragraph margin. However, we could not find any. We further removed any non-essential packages, reducing the chance that imported packages cause a change in paragraph margin. By now, our document only uses the following packages: amsmath, amsfonts, amsthm, amssymb, mathtools, multirow, booktabs, xspace, subcaption, xcolor, algorithm, algorithmic, hyperref, url. If the reviewer is aware of any of these causing issues with the paragraph margins, we are happy to find ways to replace them.
>
> > In the maze task, why does NP-SPRL have non-zero (or closer to zero) mass attributed to the unreachable center part of the maze?
>
> This behavior is caused by the uniform target distribution that SPRL aims to match in this scenario. Given that we chose a performance lower bound of 65% success rate, SPRL computes a task/goal distribution on which the agent can fulfill 65% of the tasks but that is as close as possible to the uniform target distribution. Since the uniform target distribution incorporates non-solvable tasks in the center of the maze, SPRL also encodes those tasks as long as the expected performance constraint is met. Please note that algorithms such as [ALP-GMM] or [PLR] use a similar component which amounts to randomly sampling the task/goal space with a certain probability.
>
> [ALP-GMM] Portelas, Rémy, et al. "Teacher algorithms for curriculum learning of deep rl in continuously parameterized environments." CoRL, 2020.
> [PLR] Jiang, Minqi, Edward Grefenstette, and Tim Rocktäschel. "Prioritized level replay." ICML, 2021

---

> > ### Author Response · Authors · 2021-11-20
> > **Answer to Reviewer zmCP (1/2)**
> >
> >
> > > Pick and place is solvable without a recorded demonstration with e.g. hindsight experience replay and no curriculum. It's a bit disappointing that the experiments with this environment are with a recorded trajectory
> >
> > It is true that hindsight experience replay (HER) is capable of solving these tasks without a demonstration. However, as noted in the seminal HER paper (Section 5) and discussed in the recent survey by Narvekart et al. [CRL], HER can be seen as implicitly performing a curriculum by adjusting the training tasks (in hindsight).
> > In any case, our goal was not to establish state-of-the-art performance on the Pick and Place task, but to demonstrate the shortcomings of NP-SPRL in different tasks. We chose the Pick and Place task as it is a challenging learning task for which some form of regularization is necessary. In robotics scenarios, recording human motions in order to generate a start-state curriculum may generally be a feasible approach. Consequently, we believe that it is reasonable to validate the applicability of SPRL in such a scenario.
> >
> > [CRL] Narvekar, Sanmit, et al. "Curriculum learning for reinforcement learning domains: A framework and survey." JMLR, 2020

---

### Author Response · Authors · 2021-11-20
**General Answer to the Reviewers**

We thank all reviewers for their time and feedback. We have uploaded an updated version of the paper that we believe has substantially improved thanks to the given feedback. The major changes are:

* an extended related work section
* two additional baselines: [PLR] and [VDS]
* an evaluation of WB-SPRL in the [TMA] benchmark
* an additional appendix giving intuition on the interpolation between distributions via Wasserstein Barycenters

[TMA] Romac, Clément, et al. TeachMyAgent: a Benchmark for Automatic Curriculum Learning in Deep, ICML 2021

[VDS] Zhang, Yunzhi, Pieter Abbeel, and Lerrel Pinto. "Automatic curriculum learning through value disagreement." NeurIPS, 2020.

[PLR] Jiang, Minqi, Edward Grefenstette, and Tim Rocktäschel. "Prioritized level replay." ICML, 2021

---

### Decision · Program_Chairs · 2022-01-20

**Decision:**

Reject

**Comment:**

The paper changes the metric in self-paced reinforcement learning to be a Wasserstein distance and shows that this outperforms other metrics in simple toy-like experiments.

Even after discussions with the authors, two major concerns were identified with this submission: First, the proposed modification of the metric appears to be rather incremental with regards to the original paper. Second, the proposed method is only evaluated on relatively simple environments. The approach should be evaluated on more difficult tasks.

Given that there was no strong champion for acceptance among reviewers of this paper and the above mentioned limitations, I recommend rejecting this paper.